JGP | Journal of General Physiology

Pain Focus

# Intrinsic adaptive plasticity in mouse and human sensory neurons

Lisa A. McIlvried[1]*, John Smith Del Rosario[1]*, Melanie Y. Pullen[1], Andi Wangzhou[2], Tayler D. Sheahan[1], Andrew J. Shepherd[1], Richard A. Slivicki[1], John A. Lemen[4], Theodore J. Price[2], Bryan A. Copits[1], and Robert W. Gereau IV[1,3]

**In response to changes in activity induced by environmental cues, neurons in the central nervous system undergo homeostatic plasticity to sustain overall network function during abrupt changes in synaptic strengths. Homeostatic plasticity involves changes in synaptic scaling and regulation of intrinsic excitability. Increases in spontaneous firing and excitability of sensory neurons are evident in some forms of chronic pain in animal models and human patients. However, whether mechanisms of homeostatic plasticity are engaged in sensory neurons of the peripheral nervous system (PNS) is unknown. Here, we show that sustained depolarization (induced by 24-h incubation in 30 mM KCl) induces compensatory changes that decrease the excitability of mouse and human sensory neurons without directly opposing membrane depolarization. Voltage-clamp recordings show that sustained depolarization produces no significant alteration in voltage-gated potassium currents, but a robust reduction in voltage-gated sodium currents, likely contributing to the overall decrease in neuronal excitability. The compensatory decrease in neuronal excitability and reduction in voltage-gated sodium currents reversed completely following a 24-h recovery period in a normal medium. Similar adaptive changes were not observed in response to 24 h of sustained action potential firing induced by optogenetic stimulation at 1 Hz, indicating the need for prolonged depolarization to drive engagement of this adaptive mechanism in sensory neurons. Our findings show that mouse and human sensory neurons are capable of engaging adaptive mechanisms to regulate intrinsic excitability in response to sustained depolarization in a manner similar to that described in neurons in the central nervous system.**

## Introduction

To maintain stable long-term function, neuronal networks must adapt in the face of mounting experience-dependent plasticity. Two main mechanisms of homeostatic plasticity have been proposed as master regulators of neuronal activity: synaptic scaling and homeostatic regulation of intrinsic excitability (Turrigiano et al., 1998; O'Leary et al., 2010). Synaptic scaling, which has been reported in neocortical, hippocampal, and spinal cord neurons (Lissin et al., 1998; O'Brien et al., 1998; Turrigiano et al., 1998), refers to an overall adjustment of synaptic strength that serves to compensate for perturbations in neuronal network activity. These adjustments occur in all synapses on a neuron,

scaling up or down endogenous ligand-gated receptors, thus normalizing overall neuronal firing. Alternatively, homeostatic regulation of intrinsic excitability is a non-synaptic homeostatic mechanism that controls individual cell activity by dynamically modulating intrinsic neuronal excitability, generally through changes in ion channel expression or function (Brickley et al., 2001; van Welie et al., 2004; O'Leary et al., 2010). These adaptations have been observed in excitatory-induced neuronal cells from human pluripotent stem cells, neuronal cells from cultured hippocampal and cortical mouse neurons, rodent myenteric neurons that innervate regions of

[1]Washington University Pain Center and Department of Anesthesiology, Washington University School of Medicine, St. Louis, MO, USA; [2]Department of Neuroscience and Center for Advanced Pain Studies, The University of Texas at Dallas, Dallas, TX, USA; [3]Department of Neuroscience and Department of Biomedical Engineering, Washington University School of Medicine, St. Louis, MO, USA; [4]Mid-America Transplant, St. Louis, MO, USA.

*L.A. McIlvried and J.S. Del Rosario contributed equally to this paper. Correspondence to Robert W. Gereau IV: gereaur@wustl.edu

L.A. McIlvried's current affiliation is Department of Neurobiology, University of Pittsburgh School of Medicine and Hillman Cancer Center, University of Pittsburgh Medical Center, Pittsburgh, PA, USA. T.D. Sheahan's current affiliation is Department of Cell Biology, Neurobiology, and Anatomy, Medical College of Wisconsin, Milwaukee, WI, USA. A.J. Sherpherd's current affiliation is Department of Symptom Research, Division of Internal Medicine, The University of Texas MD Anderson Cancer Center, Houston, TX, USA. This work is part of a special issue on Mapping the Pain Landscape – From Molecules to Medicine.

the enteric nervous system, tectal neurons from the *Xenopus* retinotectal circuit, and pyloric neurons from the pyloric circuit in the crab, *Cancer borealis* (Franklin et al., 1992; Desai et al., 1999; Leslie et al., 2001; Pratt and Aizenman, 2007; O'Leary et al., 2010; Zhang et al., 2018; He et al., 2020; Rue et al., 2022).

In animal models of chronic pain, changes in neuronal excitability are observed throughout the pain neuraxis, including peripheral nociceptive fibers, neurons of the dorsal horn of the spinal cord, neurons in the central nucleus of the amygdala, and pyramidal neurons in the anterior cingulate cortex (Song et al., 2003; Fu et al., 2008; Ren and Neugebauer, 2010; Weng et al., 2012; Aby et al., 2018; Li and Sheets, 2018; Miyazawa et al., 2018; Xie et al., 2018; Yang et al., 2018; Wilson et al., 2019). In addition, human sensory neurons from patients experiencing neuropathic pain also become hyperexcitable, suggesting that adaptive alterations in intrinsic neuronal excitability are present in humans (Li et al., 2017; Li et al., 2018; North et al., 2019). Thus, in both rodents and humans, sensory neurons from individuals with persistent pain show increased firing rates and changes in intrinsic excitability. This sustained increase in activity begs the question of whether sensory neurons engage homeostatic mechanisms to regulate intrinsic excitability.

In the present study, we investigated whether mechanisms of homeostatic plasticity can be engaged in small-diameter, putatively nociceptive mice and human dorsal root ganglion (DRG) sensory neurons. To drive robust changes in neuronal excitability, we incubated DRG neurons in a culture medium containing 30 mM KCl, the standard stimulus used in numerous prior studies to assess for intrinsic homeostatic plasticity in CNS neuron cultures (Leslie et al., 2001; O'Leary et al., 2010; He et al., 2020). This stimulus is not a model of pain in vitro, but a tool used to drive sustained depolarization to investigate if sensory neurons have the ability to intrinsically counteract changes in membrane potential or activity similar to CNS neurons. We found that sustained depolarization (24 h, 30 mM KCl), but not sustained activity (action potential firing), leads to adaptive alterations in neuronal excitability in mouse and human sensory neurons. The inhibition of intrinsic excitability includes changes in rheobase, input resistance, and action potential (AP) waveform. These changes were reversible within 24 h—a consistent feature of various forms of homeostatic plasticity. We also found that sustained depolarization produced only a modest alteration in voltage-gated potassium currents but a robust reduction in voltage-gated sodium currents in mouse sensory neurons. This finding suggests that voltage-gated sodium currents contribute to the decrease in neuronal excitability and potentially serve as a regulatory mechanism to drive homeostatic control in sensory neurons. Understanding the intrinsic molecular mechanisms that contribute to this adaptive downregulation of peripheral sensory neuron excitability could aid in understanding the role of these processes in restoring normal sensation in the context of injury and whether disruption of

such mechanisms might contribute to the development of chronic pain.

## Materials and methods

### Animals
All procedures were approved by the Animal Care and Use Committee of Washington University and in strict accordance with the US National Institute of Health (NIH) Guide for the Care and Use of Laboratory Animals. Adult male and female mice (7–14 wk old) utilized in experiments were housed in Washington University School of Medicine animal facilities on a 12-h light/dark cycle with access to ad libitum to food and water.

For relevant optogenetic experiments, mice were generated with conditional expression of ChR2 in TRPV1-lineage sensory neurons by crossing heterozygous TrpV1-Cre mice (provided by Mark Hoon, NIDCR [Mishra et al., 2011]) with homozygous Ai32 mice (Stock 3: 012569; The Jackson Laboratory) (Madisen et al., 2012). For the purpose of this study, we refer to these mice as "TRPV1:ChR2." These mice were previously characterized in our lab (Park et al., 2015). All other experiments were performed using a combination of C57Bl/6J and TRPV1:Ai213 (to visualize TRPV1 expressing sensory neurons with EGFP) mice bred in-house, originally obtained from The Jackson Laboratory.

### Mouse DRG cultures
Mouse DRG cultures were performed as previously described (Park et al., 2015; Samineni et al., 2017; Del Rosario et al., 2020). Briefly, mice were deeply anesthetized with isoflurane and quickly decapitated or anesthetized with an i.p. injection of ketamine (100 mg/kg) and xylazine (10 mg/kg) cocktail prior to euthanasia. After confirming a lack of response to a toe pinch, mice were perfused via the left ventricle with ice-cold Hank's buffered salt solution. DRGs were removed from all spinal segments and dissociated enzymatically with papain (0.33 mg/ml; Worthington) and collagenase type 2 (1.5 mg/ml; Sigma-Aldrich), and mechanically with trituration. Dissociated DRG were filtered (40 μM; Fisher) and cultured in DRG media (5% fetal bovine serum [Gibco] and 1% penicillin/streptomycin [Corning] in Neurobasal A medium 1x [Gibco] plus Glutamax [Life Technologies] and B27 [Gibco]) without nerve growth factor (NGF) on glass coverslips coated with poly-D-lysine and collagen.

### Human DRG cultures
Human dorsal root ganglia (hDRG) extraction, dissection, and culturing were performed as described previously (Valtcheva et al., 2016). Briefly, in collaboration with Mid-America Transplant Services, L1–L5 DRG were extracted from tissue/organ donors <2 h after aortic cross-clamp. Donor information is presented in Table 3. hDRG were placed in an *N*-methyl-D-glucamine (NMDG) solution for transport to the lab, fine dissection, and mincing. Minced DRG were dissociated enzymatically with papain (Worthington) and collagenase type 2 (Sigma-Aldrich) and mechanically with trituration. Dissociated DRG were filtered (100 μM; Fisher) and cultured in DRG media (5% fetal bovine serum [Gibco] and 1% penicillin/streptomycin [Corning] in Neurobasal A medium 1x [Gibco] plus Glutamax

[Life Technologies] and B27 [Gibco]) without NGF on coated glass cover slips.

## Assessment of intrinsic homeostatic plasticity

At DIV 3–4, mDRG cultures were treated for 24 ± 4 h with a sustained depolarizing stimulus (30 mM KCl, added to media) or sustained activity stimulus (470 nm illumination at 1 Hz, 10 ms pulses; culture plate placed on top of LED array in incubator [LIU470A; Thorlabs]) and recorded at DIV 4–5. The temperature of the media did not change after LED stimulation, indicating the LED array was not heating the media or cells. Coverslips were then changed to fresh media for 24 ± 4 h recovery and recorded. For voltage-clamp experiments, DRG neurons were tested after 24 ± 6 h in vitro. After each treatment timepoint, patch-clamp electrophysiology was performed on cultured neurons to measure the response to sustained stimulation and recovery (Fig. 1 A and Fig. 4 A). Compensatory changes in excitability in response to sustained depolarization or light stimulation were considered an indication of adaptive regulation. Human DRG cultures were treated at DIV 3–5 after glia had fallen off the neurons and exposed the cell membranes (Valtcheva et al., 2016). The investigator performing the physiology experiments was blinded to the treatment condition. Since time in culture can impact excitability (Klein et al., 1991; Delrée et al., 1993; Gold et al., 1996a; Black et al., 1999, 2018; Davidson et al., 2014), appropriate controls were included for neurons in culture for the same period as the KCl-treated neurons. Furthermore, neurons in both control and treatment groups were from the same culture preparations to account for variability in culturing between preps.

## Whole-cell patch-clamp electrophysiology

Experiments were performed on small diameter, putatively nociceptive neurons (referred to as nociceptors), cultured from mouse (<30 μm [Harper and Lawson, 1985; Gold et al., 1996a; Lawson, 2002; Fang et al., 2006; Berta et al., 2017]) and human (<60 μm [Han et al., 2015; Xu et al., 2015; Haberberger et al., 2019]) DRG. Current-clamp electrophysiological recordings were carried out in an external solution consisting of (in mM) 145 NaCl, 3 or 5 KCl, 2 CaCl$_2$, 1.2 MgCl$_2$, 7 glucose, and 10 HEPES, pH 7.3 with NaOH and ~305 mOsm. Cells were recorded within 1 h after being placed in an external solution. As an exception, acute recordings demonstrating how cells respond in culture (Fig. 1 A, bottom right, Fig. 4 B, and Fig. 5 B) were recorded in culture media as the external solution. Patch pipettes were pulled from thick-walled borosilicate glass (Sutter Instrument) using a P-97 horizontal puller (Sutter Instrument) and had resistance values of 3–4 MΩ (mouse) and 2–3 MΩ (human). The intracellular solution contained (in mM): 120 potassium gluconate, 5 NaCl, 2 MgCl$_2$, 0.1 CaCl$_2$, 10 HEPES, 1.1 EGTA, 4 Na$_2$ATP, 0.4 Na$_2$GTP, 15 sodium phosphocreatine, adjusted to pH = 7.3 with KOH, and 291 mOsm with sucrose. Fluorescence imaging and photostimulation at 470 nM were performed using an LED (M470L2; Thorlabs) coupled to the epifluorescence port of an upright microscope (BX50W1; Olympus). The LED was positioned for Köhler illumination, and light intensity was calibrated using a photodiode (S120C; Thorlabs) and power meter (PM100D; Thorlabs). Recordings were made using Patchmaster

software (Heka Instruments) controlling an EPC10 amplifier (Heka) or a MultiClamp 700B amplifier and a Digidata 1550B digitizer. All recordings were performed at room temperature with continuous, gravity-fed bath perfusion at a flow rate of 1–2 ml/min. Data were sampled at 50 kHz and analyzed off-line.

Changes in excitability were assessed in the current clamp mode by three measures: rheobase, response to suprathreshold rheobase stimuli (accommodation), and AP threshold. These were determined from a holding potential of –60 mV with a series of 1 s depolarizing step (square pulse) or ramp current injections in increments of 10 pA (mouse, cut off at 1 nA) or 100 pA (human, cut off at 10 nA). Rheobase was defined as the minimum amount of current required to evoke a single AP. Response to suprathreshold stimuli was determined by counting the number of APs evoked at one to four times rheobase. AP threshold was defined as the mV/ms rate of change of the maximum depolarization reached before AP generation. Recordings were performed with a current clamp gain of 1 pA/mV (mouse) or 10 pA/mV (human).

The passive electrophysiological properties assessed were capacitance, input resistance (R$_{in}$), and resting membrane potential (V$_{rest}$). Capacitance was determined in voltage clamp mode using the compensation of the slow capacitive transient with the amplifier circuitry. Input resistance was determined in a current clamp with a hyperpolarizing current injection of 10 pA (mouse) or 100 pA (human). V$_{rest}$ was determined in the current clamp as the average potential during a 1 s pulse with no current injection. Neurons that exhibited a V$_{rest}$ greater than –40 mV or less than –75 mV were not analyzed. The active electrophysiological properties assessed were characteristics of the AP waveform. This included AP amplitude, overshoot, rise-time, fall-time, and half-width. The first AP evoked at rheobase from a step pulse was analyzed for these characteristics. AP overshoot was measured from 0 mV to peak depolarization, and AP amplitude was measured from threshold to peak depolarization. AP rise was measured as the time from 10% to 90% of the rising phase, AP fall was measured as the time from 10% to 90% of the falling phase, and AP half-width was measured as the time from 50% of the rising phase to 50% of the falling phase (Fig. 2 C).

Voltage-clamp electrophysiological recordings to isolate potassium currents were carried out in an external solution consisting of the following components (in mM): 137 NaCl, 5 KCl, 1 MgCl$_2$, 2 CaCl$_2$, 10 HEPES, 10 glucose, 2.5 CoCl$_2$, 1 lidocaine, 0.001 tetrodotoxin (TTX). The intracellular solution contained (in mM) 140 K-gluconate, 10 HEPES, 5 EGTA, 1 MgCl$_2$, and 2 Na$_2$ATP. To isolate sodium currents, the external solution consisted of (in mM) 105 NaCl, 40 TEA-Cl, 10 HEPES, 13 glucose, 1 MgCl$_2$, and 0.1 CdCl$_2$. The intracellular solution contained (in mM) 110 CsCl, 10 EGTA, 0.1 CaCl$_2$, 4 MgATP, 0.4 Na$_2$GTP, 10 Na$_2$Phosphocreatine, 10 HEPES, and 10 TEA-Cl. To isolate the inactivating and non-inactivating components of the whole cell K$^+$ currents, we voltage-clamped the neurons at –70 mV, and the current obtained at +40 mV from a brief prepulse at –80 mV was subtracted from the current obtained after a brief prepulse at +10 mv to eliminate the non-inactivating component (Hu et al., 2003) (Fig. 6). Moreover, for sodium current recordings, mouse

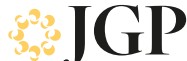

Figure 1. **Sustained depolarization of mouse DRG neurons decreased neuronal excitability. (A)** Top: Experimental design for testing homeostatic regulation of intrinsic excitability to a sustained depolarization stimulus in cultured, small diameter DRG neurons from C57Bl/6 mice. Bottom left: Viability of mouse DRG cultures after 24 h stimulation with control, 30 or 200 mM KCl treatments (*N* = 3 mice; Trypan Blue staining performed in replicate for each culture

condition). Bottom right: Example current-clamp trace from acute, 5 min application of 30 mM KCl to a mouse DRG neurons and subsequent washout of effect. KCl application resulted in a sustained depolarization to −15 ± 0.5 mV ($n$ = 5 neurons, ***P < 0.001). **(B)** Example current-clamp traces from a control cultured mouse sensory neuron (left; black) or one treated for 24 h with media containing 30 mM KCl (right; red). Current injections were delivered in ramp pulses, indicated below traces, in 10 pA increases until a reliable action potential was achieved (responders) or reaching the maximum of 1 nA without firing (non-responders; NR). **(C and D)** Comparison of the proportion of responders (C, **P = 0.0013 for comparison between control and KCl groups, **P = 0.0025 for comparison between KCl and KCl+24 h recovery groups) and rheobase from all cells (D, ***P = 0.0002) following treatment. **(E)** Example current-clamp traces from the same cultured small-diameter mouse DRG neurons as in A, with current injections delivered in step pulses. All cells responded. **(F–I)** Comparison of rheobase (F), threshold (G), input resistance (H, *P = 0.0395 for comparison between control and KCl groups, *P = 0.0262 for comparison between control + 24 h recovery and KCl +24 h recovery groups), and number of APs fired in response to one to four times rheobase current injections with example traces (I, **P = 0.0054 for comparison between control and KCl groups, ***P ≤ 0.001 for comparison between KCl and KCl + 24 h recovery groups), determined from step pulses. Treatment groups: 24 h in control, media alone ($n$ = 36; black), 24 h in media supplemented with 30 mM KCl ($n$ = 39; red), 24 h control followed by an additional 24 h in fresh media ($n$ = 35; grey), and 24 h 30 mM KCl followed by an additional 24 h "recovery" in fresh media ($n$ = 36; pink). Statistical analysis was performed with one-way and two-way ANOVAs with Tukey's multiple comparisons. Data are represented as mean ± SEM.

DRG neurons were clamped at −80 mV and a series of step pulses ranging from −60 to +20 mV were applied in +10 mV increments for 1 s (step protocol). For the ramp protocol of sodium current recording, mouse DRG neurons were held at −80 mv and ramped from −100 to 20 mV for 600 ms as shown previously (Soriano et al., 2019). Series resistance was compensated by 85% and leak subtraction was also performed for current analysis in the recordings.

## Pharmacological agents
All salts and pharmacological agents were obtained from Sigma-Aldrich, except, TTX citrate, which was obtained from Hello Bio. A 3 M stock of KCl, dissolved in DRG media with aliquots stored at −20°C until use, was diluted into culture DRG media for a final concentration of 30 mM KCl.

## RNAseq
Mouse and human DRG cultured cells were scraped from three coverslips per treatment (media, KCl, KCl +recovery) into RNAlater and frozen at −80°C until use. Samples were then thawed at room temperature, and cells were pelleted at 5,000 × $g$ for 10 min. Cells were resuspended with 1 ml of QIAzol (QIAGEN Inc.) and transferred to tissue homogenizing CKMix tubes 2 ml (Bertin Instruments). Homogenization was performed for 3 × 1 min at 20 Hz at 4°C. RNA extraction was performed with RNeasy Plus Universal Mini Kit from QIAGEN with the manufacturer-provided protocol. RNA was eluted with 30 μl of RNase-free water. Total RNAs were purified and subjected to TruSeq stranded mRNA library preparation according to the manufacturer's instructions (Illumina). The libraries were quantified by Qubit (Invitrogen) and the average size of the

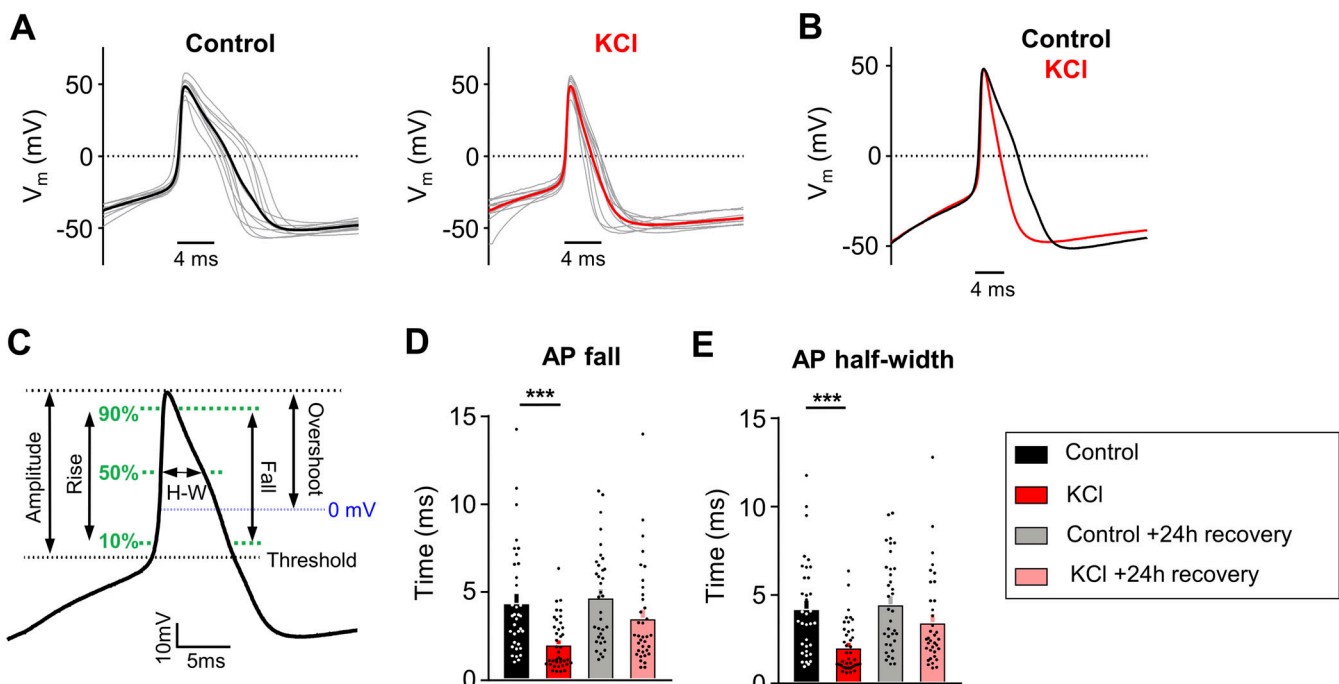

**Figure 2. AP duration is altered in mouse DRG neurons following sustained depolarization. (A)** Current-clamp AP traces from cultured mouse DRG neurons treated for 24 h with control media (left) or 30 mM KCl (right). 10 representative traces from each condition (grey) and average (bold color) are overlaid (aligned to peak depolarization) to display AP waveforms. The first AP at rheobase to step pulses was used. **(B)** Average AP traces are overlaid to display the difference with treatment (control = black, KCl = red). **(C)** Diagram of AP waveform characteristics analyzed on an average AP trace. **(D and E)** Analysis of AP fall (D, ***P = 0.0006) and half-width (E, ***P = 0.0007) from all mouse sensory neurons. Statistical analysis was performed with one-way ANOVA with Tukey's multiple comparisons. Data are represented as mean ± SEM.

libraries was determined using the High Sensitivity NGS fragment analysis kit on the Fragment Analyzer (Agilent Technologies). The normalized libraries were then sequenced on an Illumina NextSeq500 sequencing platform with 75-bp single-end reads for at least 20 million reads per sample in multiplexed sequencing experiments.

### Immunocytochemistry

Mouse DRG neurons were cultured from three mice and treated either with control media or 30 mM KCl overnight on glass coverslips coated with poly-D-lysine and collagen. The following day, cells were fixed with 10% formalin for 10 min. They were rinsed three times with 1X TBS and blocked and 10% normal goat serum (S-1000-20, NGS; vector laboratories) and 0.02% Triton-X (T-9284; Sigma-Aldrich) in TBS for 1 h. Cells were incubated with mouse anti-β III tubulin (1:1,000, Cat. 801201; BioLegend) prepared in 10% NGS for 3 h at room temperature. Cells were then rinsed three times with 1X TBS and incubated with a secondary antibody (goat anti-mouse 555, 1:2,000, A21424; Invitrogen) prepared in 10% NGS/1X TBS for 1 h at room temperature. Cells were then rinsed three times with 1X TBS and coverslips were mounted onto slides with DAPI vectashield antifade mounting medium (Cat.H-1500). Slides were stored at 4°C until imaging. Images were taken using the Leica DM6b system at 10× magnification. The mouse DRG neurons on the slides were imaged and merged using Leica Application Suite X (LAS-X, v.3.7.; Leica Microsystems).

### Data analysis

Electrophysiological data were analyzed offline using custom-written macros and the Neuromatic plug-in (Rothman and Silver, 2018) in Igor Pro (WaveMetrics) and Clampfit (Molecular Devices) and using Easy Electrophysiology software. Statistical analyses were performed with Prism software (GraphPad). Changes in excitability and passive and active properties between groups and voltage-clamp recordings were analyzed with one-way ANOVAs (Tukey multiple comparison), two-way repeated ANOVAs (Tukey multiple comparison), unpaired Student's $t$ tests, Chi squared tests, or Fisher exact $t$ test, as appropriate. Data are presented as mean ± standard error of the mean (SEM).

RNAseq Fastq files from the sequencing experiment were checked for quality by FastQC and trimming was done based on basepair-wise quality and per-base sequence content (Babraham Bioinformatics, https://www.bioinformatics.babraham.ac.uk/projects/fastqc/). Trimmed Reads were then mapped against gencode mouse genome vM16 using STAR v2.2.1 (Dobin et al., 2013). Transcripts per million (TPM) for every gene of every sample were quantified by stringtie v1.3.5 with output bam files from STAR (Pertea et al., 2015, 2016). Gene expression levels (TPM) were re-normalized against all protein-coding genes to generate the new TPM for protein-coding genes. A TPM < 0.2 was below detection level and considered not expressed. Therefore, genes were only considered when all three samples had TPMs > 0.2 in at least one of the treatment groups (media, KCl, or KCl +24 h recovery). Data were analyzed with paired student tests and with Benjamini–Hochberg multiple test

correction. For graphs, the threshold of a level of 0.2 TPM was added to all genes before taking the log transform.

### Online supplemental material

Fig. S1 shows a morphological representation and analysis of mouse DRG neurons treated with and without 30 mM KCl. Fig. S2 shows an analysis of the intrinsic excitability of mouse TRPV1:Ai213 DRG neurons treated with and without 30 mM KCl supporting data from Fig. 1. Fig. S3 shows an analysis of AP waveforms of human DRG neurons with single, delayed, and repeated firing patterns. Fig. S4 shows an assessment of global gene expression in mouse and human DRG neurons after sustained depolarization. Fig. S5 shows a representation of an unknown voltage-gated outward current in mouse DRG neurons following a voltage-step protocol.

## Results

### Sustained depolarization of mouse DRG neurons decreased neuronal excitability

To understand if homeostatic plasticity is engaged in sensory neurons, cultured DRG neurons at 3–4 days in vitro (DIV) were exposed to 30 mM KCl for 24 h to induce sustained neuronal depolarization using previous culturing protocols (Araujo and Bendhack, 2003; Robertson et al., 2014; Aguilar et al., 2017; Samineni et al., 2017; Del Rosario et al., 2020) (Fig. 1 A, top). This treatment had no impact on the viability, morphology, or size distribution of the cultured DRG neurons (Fig. 1 A, bottom left, and Fig. S1, A–C) and resulted in sustained depolarization of the cells to −15 ± 0.5 mV after 5 min or −18 ± 1.3 mV after 24 h in 30 mM KCl solution (Fig. S2 A). In all five cells where we examined the effects of the onset of KCl administration, we did not observe the generation of APs (Fig. 1 A, bottom right). Neurons treated in 30 mM KCl for 24 h recovered quickly to about −48 ± 1.7 mV on return to standard extracellular solution (Fig. S2 A).

Excitability was assessed using whole-cell patch-clamp electrophysiology after 24 ± 4 h of incubation in 30 mM KCl in culture media or after a further 24 ± 4 h recovery in fresh media (Fig. 1 A, green arrows [Rothman and Silver, 2018]). The recovery group allows us to assess for a reversal in excitability to the sustained stimulus, which is considered a hallmark indication of homeostatic plasticity (O'Leary et al., 2010). Control neurons received media changes only. Data were collected from 146 small diameter (20.6 ± 0.02 μm) DRG neurons (<30 μm, putative nociceptors [Harper and Lawson, 1985; Gold et al., 1996a; Lawson, 2002; Fang et al., 2006; Berta et al., 2017]) from eight mice (four male, four female). There were no significant differences in excitability between male and female mice, therefore data were pooled.

In current-clamp recordings (Fig. 1 B), the majority of neurons in control media fired an AP to a ramp depolarization (29/36 or 81%), and only half of the neurons fired APs to this stimulus after 24 h incubation in 30 mM KCl (19/38 or 50%; Fig. 1 C). This effect recovered within 24 h (31/36 or 86% responder rate) when neurons were placed back in fresh control media. In addition, KCl-treated DRG neurons had a significantly higher rheobase compared with the media control group, and this effect

also recovered within 24 h in fresh control media (Fig. 1 D). The neurons that did not fire any AP up to the maximum injected current ramp of 1 nA were classified as "non-responders" (NR; Fig. 1 B, right, and Fig. 1 D, top) and their rheobase was defined as 1 nA for analysis (Fig. 1 D, top of the graph). Note that a significant increase in rheobase is still evident when we analyze only responding KCl-treated neurons compared with controls (removing all non-responders) (Fig. S1 D).

We also assessed rheobase, AP threshold, and response to suprathreshold stimuli to 1-s step current injections (Fig. 1 E). We found that there were no significant changes in rheobase (Fig. 1 F) or in AP threshold (Fig. 1 G), but there was a significant change in input resistance (Fig. 1 H) using the step protocol, possibly reflecting an increase in channel insertion into the membrane. In addition, there was a significant decrease in the number of APs generated at three and four times rheobase in KCl-treated DRG neurons compared with DRG neurons treated with control media. This effect recovered within 24 h in fresh control media (Fig. 1 I). The increase in rheobase and the decrease in the percentage of DRG neurons responding to the ramp current-injection protocol as well as the decrease in input resistance and to suprathreshold step stimuli in the KCl-treated group show that there is an overall decrease in neuronal excitability of sensory neurons following a sustained depolarization. The reversibility of these effects following a 24-h washout of the depolarizing stimulus suggests that somatosensory neurons can engage a dynamic form of homeostatic plasticity.

The passive and active electrophysiological properties of the cells were also examined. There were no changes in resting membrane potential ($V_{rest}$) or whole-cell capacitance (pF) between KCl and control-treated groups (Table 1). Analysis of AP characteristics showed that KCl-treated DRG neurons lost their shoulder, which is a distinctive inflection that occurs during the descending phase of the AP (Fig. 2, A–C) (Blair and Bean, 2002; Davidson et al., 2014). This is represented by a significant decrease in AP fall time (Fig. 2 D) and a decrease in the AP half-width (Fig. 2 E). This effect partially recovered by 24 h. The AP shoulder has been attributed to calcium and sodium currents, potentially, due to the inability of these channels to completely inactivate during the AP falling phase (Blair and Bean, 2002). There was no change in AP overshoot, amplitude, or rise time (Table 1). Together, these data suggest that potential changes in voltage-gated ion channels (such as an increase in potassium channels and/or decrease in voltage-gated calcium and sodium channels) are modulating the generation and properties of APs to promote a decrease in neuronal excitability. The increase in rheobase and a decrease in the percentage of responding DRG neurons in ramp protocol but not in step protocol also suggests that adaptive changes may involve both voltage- and time-dependent gating processes.

## Prolonged exposure to 30 mM KCl alters mouse DRG neuron firing patterns

We identified three firing patterns in small DRG neurons: (1) "Single" spikers (only fire one AP at the start of current injection even with suprathreshold stimuli; also referred to as rapidly accommodating [Odem et al., 2018]), (2) "Delayed" spikers (do

not fire APs for a least 100 ms after the start of current injection of rheobase strength [average of 578 ± 73 ms]), and (3) "Repeated" spikers (fire at the start of current injection and continue firing multiple APs during the course of current injection) (Fig. 3 A). Analysis of the AP waveform from these subtypes showed a significant increase in voltage threshold, a decrease in AP amplitude, and an increase in rise time in delayed spikers as compared with single and repeated spikers (Fig. 3, B and C). Single spikers also had a significantly increased threshold, decreased AP amplitude, and decreased overshoot compared with repeated spikers (Fig. 3 C). Regardless of treatment, the majority of cells with a single spiker phenotype to step depolarizations were NR cells (Fig. 1, B and D) to ramp depolarizations (36 NRs/ 57 single spikers, or 63.2%); all of the NRs were single spikers (36 single spikers/36 NRs, or 100%).

The proportion of the three firing subtypes was significantly altered in KCl–treated cells (Fig. 3 D), such that there was a significant increase in the proportion of single spikers and a decrease in delayed and repeated spikers in KCl-treated cells compared with cells in control media (Fig. 3 D). Delayed spikers were almost absent in KCl-treated cells (1/39 or 2%). These alterations in neuronal firing after sustained depolarization were recovered to control levels in cells placed in fresh media within 24 h (Fig. 3 D). AP waveforms were analyzed by subtype to determine whether the changes in AP characteristics with KCl treatment (loss of AP shoulder quantified as decreased input resistance, AP fall time, and half-width) were due to the shift in firing patterns (i.e., an effect from just having more single firer AP waveforms) or whether KCl treatment impacted all subtype waveforms. Both single (Fig. 3, E and F) and repeated (Fig. 3, G and H) spikers showed a decrease in input resistance, AP fall time, and half-width with KCl treatment. Combined with the almost complete elimination of the delayed subtype after KCl (1/ 39 cells), these data suggest that the overall AP waveform changes with KCl treatment are due to alterations in all subtypes, in addition to the increase in the proportion of single spikers.

## Sustained activity did not alter neuronal excitability in mouse nociceptors

Exposure to 30 mM KCl produced sustained depolarization, but not persistent AP firing (Fig. 1 A, bottom right). However, the predominant phenotype seen in sensory neurons from rodents or humans with chronic pain is spontaneous firing and increased excitability, but minimal changes in membrane potential (Weng et al., 2012; Aby et al., 2018; North et al., 2019). To test whether sustained AP firing produces homeostatic changes, we used prolonged optogenetic stimulation of mouse sensory neurons (Fig. 4 A). DRG neurons were cultured from transgenic mice expressing Channelrhodopsin-2 (ChR2) in TRPV1-lineage neurons (TRPV1:ChR2 mice; see Materials and methods [Mishra et al., 2011; Madisen et al., 2012; Park et al., 2015]). To generate sustained AP firing, DRG neurons were exposed to pulsed blue light (470 nm illumination at 1 Hz, 10 ms pulses) for 24 h in a tissue culture incubator, while control TRPV1:ChR2+ DRG neurons were kept in the dark. A 1 Hz stimulus frequency was selected because it is similar to the physiological firing frequency

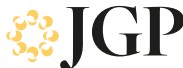

**Table 1.** Impact of sustained depolarization on additional passive and active electrophysiological properties in mouse sensory neurons, related to **Figs. 1** and **2**

| | Capacitance (pf) | V$_{rest}$ (mV) | AP amplitude (mV) | AP overshoot (mV) | AP rise (ms) |
|---|---|---|---|---|---|
| Control (*n* = 36) | 25.2 ± 2.2 | −57.5 ± 1.6 | 63.2 ± 2.3 | 47.5 ± 1.3 | 0.76 ± 0.08 |
| KCl (*n* = 39) | 20.3 ± 1.2 | −57.8 ± 1.5 | 66.3 ± 1.6 | 47.5 ± 01.1 | 0.55 ± 0.04 |
| Control + 24 h recovery (*n* = 35) | 22.5 ± 1.4 | −58.3 ± 1.4 | 66.2 ± 2.0 | 49.8 ± 1.2 | 0.73 ± 0.09 |
| KCl + 24 h recovery (*n* = 36) | 22.7 ± 1.6 | −58.0 ± 1.6 | 62.8 ± 1.5 | 48.5 ± 1.2 | 0.68 ± 1.18 |

of spontaneously active nociceptors in rodent neuropathic pain models (Djouhri et al., 2006). Additionally, stimulation at this frequency in hippocampal slices has been shown to induce homeostatic plasticity (Goold and Nicoll, 2010) and produce nociceptive behaviors in TRPV1:ChR2 mice (Grajales-Reyes et al., 2021). We confirmed that cells fire reliably to stimulation at 1 Hz (7/8 with 100% fidelity), but not to higher frequencies (e.g., 10 Hz stimulation, 0/8 with 100% fidelity, Fig. 4 B). After 24 h of 1 Hz stimulation, blue light illumination still evoked peak currents similar to those observed in control cells kept in the dark (Fig. 4 C), suggesting there was no desensitization or bleaching of ChR2 during this prolonged period of stimulation.

Excitability data were collected from 34 small diameter (19 ± 0.2 μm), ChR2-positive DRG neurons from six (three male, three female) mice. In contrast to sustained depolarization, sustained 1 Hz optogenetic activation did not significantly change the number of responders or affect rheobase during ramp protocol (Fig. 4, D–F). In addition, there were also no significant differences in rheobase, threshold, or response to suprathreshold stimuli to the step protocol (Fig. 4, G–J), thus confirming that sustained activity did not alter intrinsic excitability. We also did not find changes in input resistance (Fig. 4 K), AP fall, or half width (Fig. 4, L–O) nor in the proportion of firing pattern subtypes in DRG neurons (Fig. 4 P). There were also no changes in other AP properties or passive membrane properties (Table 2).

It is interesting to note that even under control conditions, there were significant differences (P < 0.05, Chi-squared) in the proportions of firing patterns between C57Bl/6 (Fig. 3 D "media") and TRPV1:ChR2 mice (Fig. 4 P, "dark"). There were significantly more (P < 0.05, Fisher's exact) delayed spikers in neurons from TRPV1:ChR2 mice than C57Bl/6. This raised the question of whether we did not see changes in excitability in response to sustained activity because we were recording from a different subpopulation of small diameter DRG neurons in the TRPV1:ChR2 mice that do not have adaptive plasticity mechanisms (compared to C57Bl/6). Therefore, we repeated high KCl experiments in TRPV1:ChR2 mice and found that the majority of neurons in control media fired an AP to ramp depolarization (9/11 or 81%), while fewer neurons fired APs to ramp depolarization after 24 h incubation in 30 mM KCl (2/12 or 17%) (Fig. S2 B). KCl-treated neurons had a significant increase in rheobase compared with neurons in control media in response to ramp depolarizations (Fig. S2 C). Additionally, there was an increase in rheobase (Fig. S2 D), and a significant decrease in AP fall (Fig. S2 E) and AP half-width (Fig. S2 F) in KCl-treated cells compared with neurons in control media subjected to step current injections. These

findings are very similar to our initial data on small-diameter mouse DRG neurons from C57Bl/6 mice in Fig. 1, suggesting that small-diameter DRG neurons from TRPV1:ChR2 mice also undergo intrinsic adaptive plasticity to sustained depolarization, but not activity.

## Sustained depolarization induces adaptive plasticity in human DRG neurons

We next tested whether similar adaptive mechanisms are engaged in human DRG neurons after sustained depolarization. Dorsal root ganglia were obtained from organ donors (Table 3). The experimental design was similar to that used for mice (Fig. 5 A). At 3–5 DIV (after glia migrated off the cultured neuronal membranes [Valtcheva et al., 2016]), human DRG neurons were incubated in 30 mM KCl or control media for 24 h. In contrast to mouse DRG neurons, 5/6 or 83% of human DRG neurons fired a series of APs during the rapid onset of depolarization in response to 30 mM KCl, but we did not observe sustained AP firing during the course of KCl treatment (Fig. 5 B). After this transient firing, we observed sustained depolarization to −6 ± 1 mV (*n* = 6) in human DRG neurons (Fig. 5 B, inset).

Data were collected from 125 small diameter (42 ± 1 μm) DRG neurons from nine (five male and four female) human donors (Table 3). Excitability was assessed in human DRG neurons (small to medium diameter, <60 μm [Han et al., 2015; Xu et al., 2015; Haberberger et al., 2019]) after 24 ± 4 h of incubation in 30 mM KCl in media or after a further 24 ± 4 h recovery in fresh media (Fig. 5 A, green arrows). Using ramp current injections (Fig. 5 C), we found that the percentage of KCl-treated human DRG neurons responding to current injection was not statistically significant from control media (Fig. 5 D, P = 0.09). However, similar to mice, KCl-treated human DRG neurons had a significantly increased rheobase to ramp depolarizations compared to control media, which was partially recovered within 24 h (Fig. 5 E). NR DRG neurons did not fire an AP up to the maximum ramp current of 10 nA (Fig. 5 E-top). When analyzing only the neurons that fired APs in response to ramp depolarization (that is, removing the non-responding cells), the difference in rheobase did not reach statistical significance (P = 0.056) (Fig. S3 E). While only a proportion of neurons responded to ramp protocol, all neurons responded to step current injections (Fig. 5 F). There were no significant changes in rheobase, AP threshold, and input resistance (Fig. 5, G–I), but KCl-treated human DRG neurons showed a significant decrease in the number of APs fired in response to increasing step stimuli at 4x rheobase. This change recovered after 24 h in fresh control

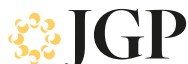

Figure 3. **Sustained depolarization alters the firing pattern of mouse DRG neurons. (A)** Example current-clamp traces of single, delayed, and repeated subtypes of firing patterns in response to suprathreshold stimuli (one to four times rheobase injections with step pulses). **(B)** AP waveform traces (average of

five representative traces per subtype; single = black, delayed = light grey, repeated = dark grey) overlaid to display differences between subtypes. The first AP at rheobase to step pulses was used. **(C)** Analysis of threshold (***P < 0.001 for comparison between single versus delayed and repeated versus delayed, *P = 0.0397 for comparison between single versus repeated), AP amplitude (*P = 0.0132 for comparison between single versus repeated, **P = 0.0044 for comparison between single versus delayed and ***P < 0.001 for comparison between repeated versus delayed) overshoot (*P = 0.0331 for comparison between single versus repeated), and AP rise (**P = 0.0023 for comparison between single versus delayed and ***P < 0.001 for comparison between repeated versus delayed) from all sensory neurons. All data were analyzed from the control treatment (single: n = 12, black; delayed: n = 5, light grey; repeated: n = 19, dark grey). Statistical analysis was performed with one-way ANOVA with Tukey's multiple comparisons. **(D)** Relative proportions of firing pattern subtypes are displayed for each treatment group (***P < 0.001, **P = 0.0018 for comparison between control and KCl groups and *P = 0.0247, ***P < 0.001 for comparison between KCl and KCl +24 h recovery groups). Statistical analysis was performed with Chi-squared and Fisher's exact t tests. **(E)** AP waveform traces (average of five representative traces per treatment; control = black, KCl = red) overlaid to display differences between treatments. The first AP at rheobase to step pulses was used. **(F)** Analysis of input resistance (*P = 0.0234), AP fall (**P = 0.0083) and AP half-width (**P = 0.0043) of single firers from control (n = 12, black) and KCl (n = 31, red) treated groups. Statistical analysis was performed with Student's t tests. **(G and H)** Same analysis as E and F, but in repeated firers (control: n = 19, KCl: n = 7, AP fall *P = 0.0433). There was only one delayed firer in the KCl-treated group, so AP analysis could not be performed for the delayed subtype. Data are represented as mean ± SEM.

media (Fig. 5 J). Therefore, the increase in rheobase-ramp and the decrease in AP firing to suprathreshold step stimuli demonstrate an overall decrease in excitability of human DRG neurons following a sustained depolarizing stimulus, similar to mouse DRG neurons.

There were no significant changes with KCl treatment on the passive membrane properties of human DRG neurons (Table 4). However, as with mouse DRG neurons, analysis of AP characteristics revealed a significant decrease in AP fall time and half-width in KCl-treated cells, which partially recovered within 24 h (Fig. 5, K–N; and Table 4). We also identified three firing patterns in human DRG neurons similar to mouse DRG neurons (Fig. S3). Analysis of the AP waveform from these subtypes showed a significant increase in voltage threshold in delayed spikers compared with single spikers and repeated spikers and a significant increase in AP rise time of delayed spikers compared with repeated spikers (Fig. S3). We did not observe significant changes in AP amplitude and AP overshoot. Additionally, there was a significant increase in the number of single spikers in KCl-treated cells, which also recovered within 24 h (Fig. 5 O). Together, this data shows that a sustained depolarizing stimulus is also capable of triggering intrinsic adaptive plasticity in human sensory neurons.

### Assessment of global gene expression in mouse and human DRG neurons following sustained depolarization

As an unbiased approach to identify potential contributors to this adaptive plasticity, we performed bulk RNAseq on mouse and human DRG cultures (n = 3 males, each) to assess global gene expression changes after 24 h treatment with 30 mM KCl (or control media) and 24 h recovery in fresh media (Table 3). A total of 21,962 genes (14,253 expressed) were analyzed in mice and 20,344 genes (14,656 expressed) in humans (see Materials and methods [Dobin et al., 2013; Pertea et al., 2015, 2016]). Hierarchical clustering showed that there were negligible alterations between treatment conditions in the global transcriptome across mouse and human DRG neurons (Fig. S4, A and B). In fact, the individual subject variance was greater than the variance caused by the treatment, with the lowest correlation still >0.86 (mouse) and 0.92 (human). Individual gene analysis showed no significant changes with treatment for human (Fig. S4 C) or mouse (Fig. S4 D) DRG neurons. A closer analysis of individual potassium and sodium channels did not reveal significant

changes between treatments as well (Fig. S4, C and D, middle and right).

### Sustained depolarization of mouse DRG neurons reduces voltage-gated sodium currents

Because no clear changes were observed in our bulk RNAseq analysis, we took a candidate-based approach to identify potential mechanisms of the observed adaptive plasticity. Voltage-gated sodium and potassium channels are the key contributors to AP generation and waveforms (Raghavan et al., 2019) and thus represent the most likely mediators of the changes in intrinsic excitability of mouse and human sensory neurons. Mouse DRG neurons were incubated in 30 mM KCl for 24 h as shown in Fig. 1 A. After 24 h, we performed voltage-clamp recordings to assess the impact of sustained depolarization on voltage-gated sodium and potassium currents.

Total voltage-gated sodium currents were significantly reduced in cells treated with KCl compared with control media when analyzed using a voltage-ramp protocol (Soriano et al., 2019) (Fig. 6, A and B) or a step-pulse protocol (Fig. 6, C–E). When KCl-treated neurons were returned to control media for 24 h, the decrease in sodium channel currents was recovered (Fig. 6 B). The notably larger reduction in sodium currents measured in the ramp protocol compared with the step protocol suggests that a time-dependent property in sodium channel gating might contribute to the reduction in excitability. This is consistent with the data shown in Fig. 1, B and E, where ramp depolarizations revealed more substantial differences after prolonged KCl treatment compared with step depolarization in the current clamp experiments.

We also observed an unidentified outward current that was slightly increased in KCl-treated cells during the isolation of sodium currents in the step-pulse protocol. This unblocked outward current cannot be attributed to potassium since potassium ions were omitted from the intra- and extracellular solutions (Fig. S5). This may be a voltage-gated sodium channel–derived outward current as previously described in rat and human DRG neurons while isolating voltage-gated $Ca^{2+}$ currents (Hartung et al., 2022).

We further investigated if there were changes in current density from the two main types of potassium currents present in DRG neurons: a fast inactivating (A-type) and a non-inactivating component (Gold et al., 1996b; Hu et al., 2003).

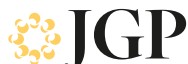

Figure 4. **Sustained activity did not alter neuronal excitability in mouse nociceptors. (A)** Experimental design for testing homeostatic regulation of intrinsic excitability to sustained activity in cultured small DRG neurons from TRPV1:ChR2 transgenic mice. **(B)** Left: Survival curve of nociceptors exposed to

470 nm blue illumination at 1 or 10 Hz, 10 ms pulses. At 1 min, 7/8 cells were still reliably firing with 1 Hz stimulation. Right: Example trace of a cell firing at 1 Hz. **(C)** A 1-s pulse of continuous illumination evoked similar peak currents in control cells kept in the dark (*n* = 16; black) and cells stimulated with 24 h illumination at 1 Hz on an LED array (*n* = 18; blue). Excitability was analyzed as described in Fig. 1, but with treatment groups to test a sustained activity stimulus. **(D–F)** Comparison of example current clamp traces (D), the proportion of responders (E), and rheobase (F) determined from ramp current injections. **(G–K)** Comparison of example current-clamp traces (G), rheobase (H), threshold (I), response to suprathreshold stimuli (J), and input resistance (K) determined from step current injections. Treatment groups: cells kept in the dark (*n* = 16; black), cells exposed to 24-h illumination at 1 Hz (*n* = 18; blue). All cells were cultured from TRPV1:ChR2 mice. **(L)** Current clamp AP traces from nociceptors treated for 24 h in the dark (left) or 1 Hz stimulation (right). Eight representative action potential traces from each condition (grey) and average (bold color) are overlaid. The first AP at rheobase to step pulses was used. **(M)** Average AP traces are overlaid to show similar AP waveforms. **(N and O)** Analysis of AP fall (N) and half-width (O) from all nociceptors. **(P)** Relative proportions of firing pattern subtypes are displayed for each treatment group. Data are represented as mean ± SEM.

## Discussion

We isolated both currents and compared the peak and steady current density in KCl and control-media neurons (see Materials and methods for description). There was no difference in the non-inactivating component of whole-cell potassium currents, and while there was a trend toward an increase in A-type current density in the KCl-treated neurons, this was not statistically significant (Fig. 6, F–H).

The cellular and molecular mechanisms that dictate Hebbian and homeostatic plasticity have been widely studied in the CNS due to their critical role in maintaining the efficiency of neuronal communication in the event of global perturbation of the overall synaptic network (Turrigiano et al., 1998; Desai et al., 1999; Malenka and Bear, 2004; Gómez et al., 2021). However, whether similar modulatory mechanisms are present in peripheral sensory neurons has been largely unexplored. Here, we first used a strong, sustained depolarizing stimulus (30 mM KCl for 24 h, as has been described in previous studies of homeostatic plasticity [Moulder et al., 2003; O'Leary et al., 2010]) to address this question in mouse and human sensory neurons from DRG. The choice of this stimulus was not based on a desire to mimic physiological or pathological pain conditions in culture, but rather to understand how sensory neurons respond to persistent changes in membrane potential. Using 30 mM KCl as a tool, we aimed to investigate whether sensory neurons have the ability to adaptively regulate intrinsic excitability similarly to CNS neurons.

We found that small-diameter mouse DRG neurons showed significant inhibition of intrinsic excitability following prolonged depolarization, but not prolonged AP firing (Figs. 1 and 4). In our study, chronically depolarized mouse DRG neurons showed an increase in rheobase, a decrease in input resistance, and a substantial decrease in AP generation to suprathreshold current injections to up to four times rheobase compared with control-treated neurons.

These findings are similar to reports of chronic depolarization of hippocampal and cortical neurons in culture using a high concentration of KCl (Leslie et al., 2001; O'Leary et al., 2010). In both the present study and the report by O'Leary et al., prolonged depolarization induced by a high KCl culture medium led to increased rheobase and decreased input resistance that generated an overall reduction in intrinsic excitability. In hippocampal neurons, this was additionally accompanied by a hyperpolarization of membrane potential, suggesting the presence of homeostatic mechanisms that regulate membrane potential (O'Leary et al., 2010). Thus, the inciting stimulus, depolarization, is directly opposed in some way to generate a compensatory hyperpolarization of membrane potential that is evident in the removal of the inciting stimulus. In sensory neurons, this type of homeostatic mechanism regulating membrane potential was not apparent, and as such we are opting to refer to the changes we are observing as a form of adaptive plasticity, rather than homeostatic plasticity. We do note that the changes we observe in excitability overlap to a large extent with observations of homeostatic plasticity reported in CNS neurons. The only notable difference is that there does not seem to be a direct homeostatic mechanism that opposes the depolarization induced by high-KCl medium in sensory neurons.

Overall, depolarization of sensory neurons generates a hypoexcitable state that reverses on removing the inciting stimulus. This suggests that there are both similarities and differences in the mechanisms of homeostatic regulation of intrinsic excitability in sensory neurons compared with the CNS cell types discussed above. In both studies, the parameters of membrane excitability altered by sustained depolarization partially or fully returned to baseline within 24–48 h. These findings show that mouse DRG neurons can exert adaptive control to regulate their intrinsic excitability similar to neurons in the CNS.

Adaptive plasticity was also engaged in human DRG neurons chronically depolarized with KCl (Fig. 5), suggesting that this plasticity is conserved across species. Homeostatic plasticity has been observed in multiple species, including flies (Frank, 2014), crustaceans (Turrigiano et al., 1994; Golowasch et al., 1999), and rodents (O'Leary et al., 2010). To our knowledge, our study is the first to provide evidence for a similar form of adaptive plasticity in primary human sensory neurons. In addition, we found that

**Table 2. Impact of sustained activity on additional passive and active electrophysiological properties in mouse nociceptors, related to Fig. 4**

|  | Capacitance (pf) | V_rest (mV) | AP amplitude (mV) | AP overshoot (mV) | AP rise (ms) |
| --- | --- | --- | --- | --- | --- |
| Control (*n* = 16) | 25.3 ± 2.0 | −51.4 ± 1.2 | 51.7 ± 3.1 | 42.4 ± 2.3 | 1.1 ± 0.1 |
| 1 Hz (*n* = 18) | 29.2 ± 3.2 | −53.0 ± 1.0 | 46.0 ± 2.6 | 36.2 ± 2.2 | 1.0 ± 0.1 |

**Table 3. Human DRG donor characteristics, related to Figs. 5 and S2**

| Age | Sex | Race | BMI | Cause of death (COD) | Ephys | RNAseq |
|---|---|---|---|---|---|---|
| 18 | F | White | 26 | Head trauma/MVA | X | |
| 18 | M | White | 23 | Head trauma/GSW | X | |
| 5 | M | White | 18 | Anoxia/seizure | X | |
| 34 | F | Hispanic | 20 | Anoxia/seizure | X | |
| 25 | F | Black | 21 | Head trauma/MVA | X | |
| 34 | M | White | 26 | Anoxia/OD | X | X |
| 34 | M | White | 22 | Head trauma/MVA | X | X |
| 11 | M | White | 16 | Head trauma/GSW | | X |
| 20 | F | White | 27 | Anoxia/OD | X | |
| 6 | M | White | 17 | Head trauma/MVA | X | |

the firing patterns of both mouse and human DRG neurons were significantly altered after sustained depolarization. Our results showed three main types of firing patterns: single spikers (also referred to as rapidly accommodating), delayed spikers (also referred to as non-accommodating [Odem et al., 2018]), and repetitive spikers (Figs. 3 and S3). Sustained depolarization shifted the firing pattern of both mouse and human sensory DRG neurons to preferentially single spikers (Fig. 3 D and Fig. 5 O).

Treatment with elevated concentrations of extracellular KCl resulted in sustained depolarization but not sustained activity of DRG neurons in culture, similar to observations from cultured CNS neurons (Leslie et al., 2001; Moulder et al., 2003; O'Leary et al., 2010). We, therefore, studied whether similar homeostatic mechanisms are engaged in DRG neurons in response to cell-autonomous increases in activity. Using an optogenetic approach, we photostimulated TRPV1-lineage nociceptors at a sustained rate of 1 Hz for 24 h and measured whether these DRG neurons showed changes in intrinsic excitability after prolonged firing, as shown previously in hippocampal pyramidal neurons (Goold and Nicoll, 2010). We found that a sustained increase in AP firing was not sufficient to trigger homeostatic regulation of intrinsic excitability in sensory neurons (Fig. 4). While persistent firing at 1 Hz for 24 h produces some degree of net depolarization, it is clear that this is insufficient to engage the adaptive mechanisms observed with sustained depolarization in sensory neurons. This was surprising because pharmacological, electrical, and optogenetic stimuli have been used to study activity-induced homeostatic plasticity in hippocampal and neocortical networks (Franklin et al., 1992; Desai et al., 1999; Brickley et al., 2001; van Welie et al., 2004; Goold and Nicoll, 2010; O'Leary et al., 2010; Fong et al., 2015). Our data suggest that chronic AP activity is not sufficient to induce homeostatic scaling of intrinsic voltage-gated ion channels to adjust neuronal excitability in peripheral sensory neurons. It is possible that the absence of homeostatic plasticity in our sustained activity experiments could be due to inadequate stimulation parameters in these experiments. First, stimulation for >24 h may be required. A previous study found that complete Freund's adjuvant (CFA; an inflammatory stimulus) injected into the hind paw of rodents decreased evoked AP firing frequencies of nociceptors at 8 wk

after injection but not at 2 days (Weyer et al., 2016), suggesting that >48 h of activity may be required to induce plasticity in this model. Second, DRG neurons fire in multiple patterns to different stimuli and may require a different pattern of input or activity to engage homeostatic mechanisms.

Our data show that human and mouse sensory neurons exert intrinsic adaptive control in response to sustained depolarization. What mechanism of action is responsible for driving this adaptive plasticity in sensory neurons? Voltage clamp recordings showed that sustained depolarization of mouse DRG neurons elicits a profound decrease in voltage-gated sodium current density compared with control neurons (Fig. 6), similar to the changes observed in hippocampal neurons (O'Leary et al., 2010). However, we did not observe statistically significant changes in voltage-gated potassium current density, which further supports the critical role of sodium channels in the regulation of excitability induced by chronic depolarization and their crucial role in the adaptive response in DRG neurons.

Interestingly, sustained activation of sodium channels in cultured neurons or sustained elevation of intracellular calcium concentration in cultured adrenal chromaffin cells induces endocytosis and degradation of sodium channels, as well as destabilization of sodium channel mRNA (Wada et al., 2004; Cusdin et al., 2008). These processes involve the activation of calpain, calcineurin, and protein kinase C. Whether similar mechanisms are engaged in sensory neurons during adaptive plasticity will demand additional experiments to define the underlying mechanisms of this reduction of sodium currents in sensory neurons produced by sustained depolarization.

We cannot exclude the potential involvement of other ion channels such as voltage-gated calcium channels, calcium-activated potassium channels, or various leak currents involved in the regulation of the process of adaptive plasticity. Indeed, we noted a significant reduction in input resistance in mouse, but not human sensory neurons, and an increase in an unidentified outward current in neurons that underwent sustained depolarization compared with controls in our mouse voltage clamp experiments. These factors could contribute to the overall reduction in excitability and will require a more detailed investigation in future studies. However, the robust reduction of

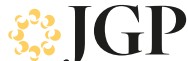

Figure 5. **Sustained depolarization induces adaptive plasticity in human DRG neurons. (A)** Experimental design for testing homeostatic regulation of intrinsic excitability to a sustained depolarization stimulus in cultured, small/medium diameter DRG neurons from human tissue/organ donors. **(B)** Left: example current-clamp trace from acute, 5 min application of 30 mM KCl to a human nociceptor and subsequent washout of effect. Right: KCl application resulted in a sustained depolarization to −6 ± 1 mV ($n$ = 6). Excitability was analyzed as described in Fig. 1, but with data collected from human sensory neurons.

**(C–E)** Comparison of example current clamp traces (C), the proportion of responders (D), and rheobase (E) determined from ramp current injections (*P = 0.0256). **(F–J)** Comparison of example current-clamp traces (F), rheobase (G), threshold (H, **P = 0.0073), input resistance (I), and response to suprathreshold stimuli determined from step current injections (J, **P = 0.0077 for comparison between control and KCl groups, and *P = 0.0248, ***P < 0.001 for comparison between KCl and KCl +24 h recovery groups). Treatment groups: 24 h in control, media alone (*n* = 36; black), 24 h in media supplemented with 30 mM KCl (*n* = 34; red), 24 h control followed by an additional 24 h in fresh media (*n* = 27; grey), and 24 h 30 mM KCl followed by an additional 24 h "recovery" in fresh media (*n* = 28; pink). **(K)** Current-clamp AP traces from cultured human DRG neurons treated for 24 h with control media (left) or 30 mM KCl (right). 10 representative traces from each condition (grey) and average (bold color) are overlaid to display AP waveforms. The first AP at rheobase to step pulses was used. **(L)** Average AP traces are overlaid to display differences with treatment (control = black, KCl = red). **(M and N)** Analysis of AP fall (M, **P = 0.0027) and half-width (N, **P = 0.0021) from all DRG neurons. **(O)** Relative proportions of firing pattern subtypes are displayed for each treatment group (**P = 0.0016 for comparison between control and KCl groups and *P = 0.0209 for comparison between KCl and KCl +24 h recovery groups). Statistical analysis was performed with one-way and two-way ANOVAs with Tukey's multiple comparisons. Chi-squared and Fisher's exact tests were also performed. Data are represented as mean ± SEM.

sodium currents is likely to contribute substantially to the overall inhibition of neuronal excitability in DRG neurons we observed here.

These data also expand our understanding of the extent to which different neuron types, from either the peripheral nervous system or CNS, utilize different intrinsic mechanisms to drive homeostatic plasticity. In cortical and hippocampal neurons, synaptic scaling of AMPA and NMDA receptors as well as changes in intrinsic excitability via voltage-gated calcium channels contribute to homeostatic mechanisms during periods of chronic disturbances in synaptic activity (Turrigiano et al., 1998; Watt et al., 2000; Goold and Nicoll, 2010; O'Leary et al., 2010). Here, we show that DRG neurons appear to employ alterations in voltage-gated sodium channels to modulate intrinsic excitability during periods of sustained depolarization. DRG neurons express at least five different types of sodium channels (Na$_v$s) (Ho and O'Leary, 2011; Griffith et al., 2019) with Na$_v$1.7, Na$_v$1.8, and Na$_v$1.9 representing the predominant subtypes (Yu and Catterall, 2003). In addition, Na$_v$1.7, Na$_v$1.8, and Na$_v$1.9 are preferentially expressed in small-diameter putative nociceptors and are known to be linked to human pain syndromes (Ho and O'Leary, 2011; Emery et al., 2016). Na$_v$1.7 is localized primarily in axons, where it promotes the initiation and conduction of AP (Toledo-Aral et al., 1997; Yu and Catterall, 2003; Emery et al., 2016). Na$_v$1.8 and Na$_v$1.9 show a more restricted expression in sensory neurons, where they produce the majority of the current responsible for the depolarizing phase of the APs (a critical step for the generation of multiple APs during ongoing neuronal activity) and the persistent TTX-resistant current found in small DRG neurons, respectively (Blair and Bean, 2002; Choi et al., 2007; Ho and O'Leary, 2011). Our findings reveal that the total voltage-gated sodium current is reduced after

chronic depolarization. If specific sodium channel subtypes are altered during adaptive plasticity, these could represent important therapeutic targets for human chronic pain conditions.

Our RNAseq data revealed no significant changes in the expression of individual potassium and sodium channel genes in KCl-treated groups compared with controls (Fig. S4). We can only conclude that at the RNA level, as detectable using bulk RNA sequencing, there are no significant changes in gene expression 24 h after prolonged depolarization. We cannot exclude the possibility of altered trafficking, internalization, or post-translational modifications of voltage-gated sodium channels in producing the observed reductions in sodium current density. Moreover, it is important to note that RNAseq was performed on DRG cultures that contain glia and other non-neuronal cells. This is a potential confound that could mask changes in RNA expression in nociceptors. Single-cell sequencing or traditional quantitative PCR approaches would be needed to specifically address this issue.

Our data support the concept that homeostatic plasticity is not restricted to neurons in the CNS but can also be engaged in peripheral somatosensory neurons. Whether these adaptive changes are engaged in the context of physiological or pathophysiological activation of sensory neurons is not known. Recent studies suggest that this may indeed be the case. In both rodent and human sensory neurons, prolonged exposure to inflammatory mediators has been shown to induce a similar suppression of neuronal excitability. For example, the proinflammatory cytokine macrophage migration inhibitory factor (MIF) depolarized and drove activity in putative nociceptors and transitioned neurons from a hyperexcitable state (repetitive spiking phenotype) following acute stimulation to a hypoexcitable state (single spiking phenotype) following prolonged exposure in rodents

Table 4.   **Impact of sustained depolarization on additional passive and active electrophysiological properties of human DRG neurons, related to Fig. 5**

| | Capacitance (pf) | RMP (mV) | AP amplitude (mV) | AP overshoot (mV) | AP rise (ms) |
|---|---|---|---|---|---|
| Control (*n* = 36) | 123 ± 11 | −58 ± 1 | 76 ± 1 | 60 ± 1 | 0.53 ± 0.04 |
| KCl (*n* = 34) | 118 ± 13 | −60 ± 1 | 72 ± 3 | 52 ± 2[a] | 0.52 ± 0.05 |
| Control + 24 h recovery (*n* = 27) | 120 ± 8 | −60 ± 1 | 77 ± 1 | 59 ± 1 | 0.47 ± 0.04 |
| KCl +24 h recovery (*n* = 28) | 131 ± 10 | −58 ± 1 | 73 ± 2 | 59 ± 2[a] | 0.49 ± 0.03 |

AP overshoot was statistically significant when comparing KCl group versus control + 24 h recovery and KCl group versus KCl + 24 h recovery.

[a]P = 0.0357, one-way ANOVA with Tukey's multiple comparisons. Data are represented as mean ± SEM.

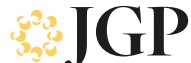

Figure 6. **Sustained depolarization of mouse DRG neurons inhibits voltage-gated sodium channel currents. (A)** Example traces of voltage-gated sodium currents from control-treated cells (black) and KCl-treated cells (red) using a voltage-ramp protocol are shown below traces. Cells were held at −80 mv and ramped from −100 to +20 mV for 600 ms. **(B)** Statistical analysis of control cells (black, n = 28), KCl-treated cells (red, n = 31), and cells treated with additional fresh media for a 24 h-"recovery" period: (grey, n = 8), (pink, n = 9), ***P < 0.001 for comparison between control and KCl groups, *P = 0.0488 for comparison between KCl and KCl +24 h recovery groups. **(C)** Example traces of voltage-gated sodium currents from control-treated cells using a voltage-step pulse protocol shown below traces. **(D)** I/V curve of voltage-gated sodium currents from control and KCl-treated cells. Cells were held −80 mV and a series of step pulses ranging from −60 to +20 mV were applied in +10 mV increments for 1 s. **(E)** Current density analysis of control cells (black, n = 29) versus KCl-treated cells (red, n = 31) at 0 mV, **P = 0.0089. **(F)** Example traces of A-type and non-inactivating voltage-gated potassium currents in mouse DRG neurons using a voltage-step pulse protocol are shown below traces. Cells were clamped at −70 mV and the current obtained at +40 mV from a brief prepulse at −80 mV was subtracted from the current obtained after a brief prepulse at +10 mv to eliminate the non-inactivating component. The isolation protocol of A-type voltage-gated potassium currents is shown below the representative traces. **(G and H)** Current density analysis of A-type currents and non-inactivating currents (control-black, KCl–red). Statistical significance was calculated with the Mann-Whitney test or Student's t test and Kruskal–Wallis test with Dunn's test for multiple comparisons. Scatter plots and mean ± SEM for current amplitudes.

(Bavencoffe et al., 2022). In addition, we recently found similar results in human DRG neurons in culture, where acute exposure to the inflammatory mediator bradykinin leads to hyperexcitability, while more prolonged exposure leads to hypoexcitability (Yi et al., 2024).

Adaptive changes in firing patterns of peripheral sensory neurons do not happen in isolation, but rather are likely to impact overall function in the pain neuraxis. For example, in vivo recordings of DRG neurons using calcium imaging in models of chronic pain in mice suggest that sensory neurons synchronize their activity within hours after injury and that this synchrony is necessary to drive cortical plasticity in somatosensory cortex (Chen et al., 2023). Thus, adaptive changes in sensory neuron excitability appear to have

implications for CNS circuit adaptations that are related to pain chronification.

These findings raise a number of questions regarding the relationship between the adaptive plasticity of peripheral sensory neurons and the mechanisms responsible for driving the transition from acute to chronic pain. If nociceptors are capable of engaging adaptive mechanisms that dampen intrinsic excitability, why are these mechanisms not engaged in the context of painful conditions associated with chronic nociceptor activity? Is this adaptive plasticity lost or disrupted during the transition to chronic pain, and further, can variability in the efficacy of adaptive plasticity explain why some recover from injury, while others transition to chronic pain? These questions may be answered in future studies using animal models of chronic pain and sensory neurons from human donors with a history of chronic pain.

### Data availability

The data are available from the corresponding author upon reasonable request. Data from figures and supplementary figures are also openly available in a public repository, http://doi.org/10.17632/tmp5ry5sz8.3, McIlvried et al. (2024).

## Acknowledgments

David A. Eisner served as editor.

We thank the organ donors, the families, and Mid-America Transplant Services for the valuable gifts that made this research possible. We acknowledge the Genome Center and Yeunhee Kim, Ph.D. in UTD Research Core Facilities for mRNA library preparation and sequencing. We also thank Sherri Vogt for the maintenance of animal colonies, and Judith Golden, Ph.D. for assistance with manuscript editing.

This research was supported by the National Institute Of Neurological Disorders And Stroke of the National Institutes of Health (NIH) through the PRECISION Human Pain Network (RRID:SCR_025458), part of the NIH HEAL Initiative (https://heal.nih.gov/) under award number U19NS130607 to R.W. Gereau and B.A. Copits, and award number U19NS130608 to T.J. Price. This study was also supported by NIH grants DA007261 (to L.A. McIlvried), NS113422 (to J.S. Del Rosario), NS065926 (to T.J. Price), and NS042595 (to R.W. Gereau).

Author contributions: L.A. McIlvried: Conceptualization, Formal analysis, Funding acquisition, Investigation, Methodology, Project administration, Validation, Visualization, Writing - original draft, Writing - review & editing, J.S. Del Rosario: Formal analysis, Funding acquisition, Investigation, Methodology, Validation, Visualization, Writing - original draft, Writing - review & editing, M.Y. Pullen: Conceptualization, Investigation, Methodology, A. Wangzhou: Data curation, Formal analysis, Software, T.D. Sheahan: Resources, Writing - review & editing, R.A. Slivicki: Resources, J.A. Lemen: Resources, T.J. Price: Formal analysis, Project administration, Supervision, Writing - review & editing, B.A. Copits: Conceptualization, Formal analysis, Funding acquisition, Investigation, Methodology, Project administration, Supervision, Visualization, Writing - review & editing, R.W.

Gereau, IV: Conceptualization, Funding acquisition, Methodology, Project administration, Supervision, Visualization, Writing - review & editing.

Disclosures: The authors declare no competing interests exist.

Submitted: 23 September 2023

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

**Supplemental material**

**A** **Control Media**

**B** **KCl**

**C**

**D** **Rheobase-ramp**

Figure S1.   **Analysis of mouse sensory neurons treated with and without 30 mM KCl, related to** Fig. 1. **(A and B)** Representative images of control media-treated DRG neurons (left) and KCl-treated DRG neurons (right). β-III tubulin antibody (red) was used to visualize mouse sensory neurons. **(C)** Relative frequency percentage of DRG neurons treated with control media ($n$ = 3,810) and 30 mM KCl ($n$ = 3,069). **(D)** Analysis of rheobase-ramp of responding DRG neurons treated with control media and KCl (**P = 0.0032). One-way ANOVAs with Tukey's multiple comparisons. Data are represented as mean ± SEM.

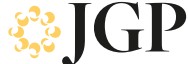

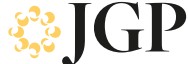

Figure S2. **Analysis of mouse TRPV1:Ai213 sensory neurons treated with and without 30 mM KCl, related to** Fig. 1. **(A)** Analysis of resting membrane potential of sensory neurons after 24 h in 30 mM KCl and recorded in 30 mM KCl solution ($n = 11$, ***$P < 0.001$, paired $t$ test). **(B)** Comparison of the proportion of responders (B, **$P = 0.0033$, Fisher's exact test). **(C–F)** Analysis of rheobase-ramp (C, **$P = 0.0052$, Mann-Whitney test), rheobase-step (D, *$P = 0.0177$, Mann-Whitney test), AP fall (E, ***$P = 0.002$, Mann-Whitney test), and AP half-width (F, **$P = 0.0026$, Student's $t$ test). Data are represented as mean ± SEM.

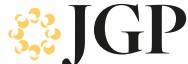

Figure S3.   **Analysis of active and passive properties of human DRG sensory neurons, related to** Fig. 5. **(A–D)** Analysis of AP threshold (*P = 0.0232 for single versus delayed and ***P < 0.001 for delayed versus repeated), AP amplitude, AP overshoot, and AP rise (**P = 0.006 for delayed versus repeated) from all human sensory neurons. All data were analyzed from the control treatment (single: *n* = 9, black; delayed: *n* = 7, light grey; repeated: *n* = 20, dark grey). **(E)** Rheobase-ramp comparing responding neurons from control media (black, *n* = 32) versus KCl-treated cells (red, *n* = 24), P = 0.0558 using Mann-Whitney test. Statistical analysis was performed with one-way ANOVAs with Tukey's multiple comparisons or Kruskal–Wallis test with Dunn's multiple comparisons. Data are represented as mean ± SEM.

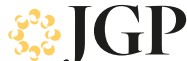

**Figure S4.** **Assessment of global gene expression in mouse and human DRG neurons following sustained depolarization. (A and B)** Hierarchical clustering of RNA-seq samples from human and mouse DRG sensory neurons. *N* = 3 cultures each, denoted by #1–3. Neurons were collected after ~4 days in vitro and the following treatment groups were studied: Media (control), 24 h KCl (24 h cultured in KCl media), and 24 h Rec (additional 24 h recovery in fresh media after cultured in KCl for 24 h). **(C and D)** Differential gene expression analysis between condition with KCl versus media fold change (X-axis) and 24 h Rec versus media (Y-axis). Middle and right panels: Isolated potassium and sodium ion channels for better representation.

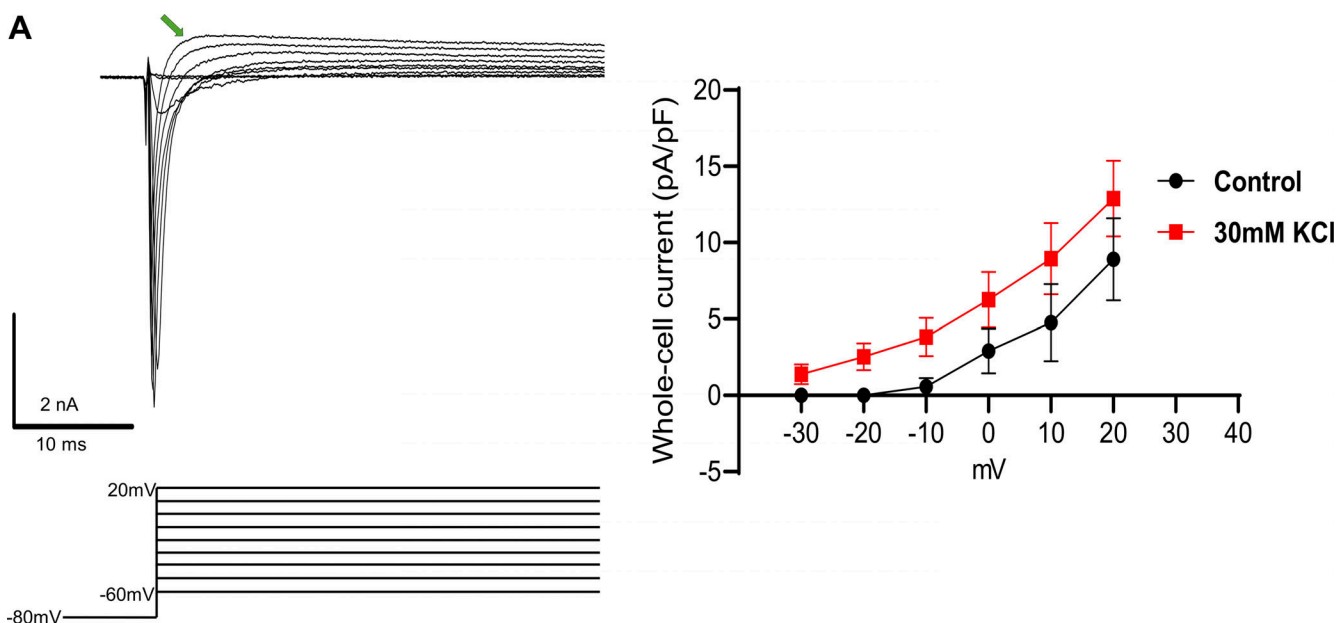

Figure S5. **Mouse DRG neurons exhibit an unknown voltage-gated outward current, related to** Fig. 6. **(A)** Example traces of voltage-gated outward currents in mouse DRG neurons using a voltage-step pulse protocol are shown below traces. Current-density analysis of an unknown outward current from control (right; black) and KCl-treated cells (right; red). Analysis was done using the value at the peak of the outward current before the current decay (green arrow).

