## [Peer Review File · The Journal of General Physiology]

Intrinsic Adaptive Plasticity in Mouse and Human Sensory Neurons

Robert Gereau, Lisa McIlvried, John Del Rosario, Melanie Pullen, Andi Wangzhou, Tayler Sheahan, Andrew Shepherd, Richard Slivicki, John Lemen, Theodore Price, and Bryan Copits

Corresponding Author(s): Robert Gereau, Washington University in St. Louis School of Medicine and Andrew Shepherd, Washington University in St. Louis School of Medicine

Review Timeline:

Submission Date:	September 23, 2023
Editorial Decision:	October 25, 2023
Revision Received:	June 7, 2024
Editorial Decision:	July 3, 2024
Revision Received:	September 20, 2024
Editorial Decision:	September 24, 2024
Revision Received:	November 23, 2024

Editor: David Eisner

Transaction Report:

DOI: <https://doi.org/10.1085/jgp.202313488>

Date: 10-25-2023 10:12:39
From: jgp@msubmit.net
To: gereaur@wustl.edu
Subject: Decision on JGP submission 202313488-Gereau IV

October 25, 2023

Dr. Robert W Gereau IV
Washington University in St. Louis School of Medicine
Anesthesiology
660 S Euclid Ave
Campus Box 8054
St Louis, MO 63110

Re: 202313488

Dear Dr. Gereau,

Thank you for submitting your manuscript, entitled "Intrinsic Homeostatic Plasticity in Mouse and Human Sensory Neurons" to JGP. Your manuscript has now been seen by 3 reviewers, whose comments are appended below. You will see that the reviewers have raised several concerns that should be addressed prior to further consideration of the manuscript at JGP. In particular, I would identify the following. (1) The differences between the results of optogenetic stimulation and 30 mM KCl. The ms suggests that the former may not represent physiological activity. While this may be the case, the reviewers mention that 30 mM KCl is certainly not a simple model of increased electrical activity. At the very least, this point should be discussed. Furthermore, it seems essential to know what effect 30 mM KCl would have in the TRPV1 neurons used for the optogenetic experiments. In this context please note the comments of reviewer #3 about the different populations of neurons. (2) As pointed out by reviewer #2, there is a need to show reversibility of the effect of 30 mM KCl on the sodium current and that reviewer makes other useful suggestions to probe mechanism.

We would be pleased to receive a suitably revised manuscript that addresses these concerns, which will be re-reviewed, most likely by some or all of the original referees. Based on the scope of the requested changes, we typically anticipate that the revision process will take no longer than 6 months, however, we understand you may need additional time to work on your resubmission to JGP. We therefore ask that you simply keep us informed as to a realistic submission timeline that is appropriate for your particular circumstances. In addition, please do not hesitate to contact me (via the editorial office) if you feel that a discussion of the reviewers' and editors' comments would be helpful.

Please submit your revised manuscript via the link below along with a point-by-point letter that details your responses to the editors' and reviewers' comments, as well as a copy of the text with alterations highlighted (boldfaced or underlined). If the article is eventually accepted, it would include a 'revised date' as well as submitted and accepted dates. If we do not receive the revised manuscript within one year, we will regard the article as having been withdrawn. We would be willing to receive a revision of the manuscript at a later time, but the manuscript will then be treated as a new submission, with a new manuscript number.

Please pay particular attention to recent changes to our instructions to authors in the following sections: Data presentation, Blinding and randomization and Statistical analysis, under Materials and Methods, as shown here: <https://rupress.org/jgp/pages/submission-guidelines#prepare>. Re-review will be contingent on inclusion of the required information (including for data added during revision) and demonstration of the experimental reproducibility of the results. Also, to improve the reproducibility of published content, we have partnered with SciScore. Authors are prompted in eJP to copy and paste the Materials and Methods section of their manuscript for a SciScore assessment when submitting their revised manuscript. Authors are encouraged (not required) to further revise their Materials and Methods if the SciScore is below 4. More information can be found here: <https://rupress.org/jgp/pages/submission-guidelines#sciscore>

Please note, JGP now requires authors to submit Source Data used to generate figures containing gels and Western blots with all revised manuscripts (when applicable). This Source Data consists of fully uncropped and unprocessed images for each gel/blot displayed in the main and supplemental figures. If your paper includes cropped gel and/or blot images, please be sure to provide one Source Data file for each figure that contains gels and/or blots along with your revised manuscript files. File names for Source Data figures should be alphanumeric without any spaces or special characters (i.e., SourceDataF#, where F# refers to the associated main figure number or SourceDataFS# for those associated with Supplementary figures). The lanes of the gels/blots should be labeled as they are in the associated figure, the place where cropping was applied should be marked (with a box), and molecular weight/size standards should be labeled wherever possible. Source Data files will be made available to reviewers during evaluation of revised manuscripts and, if your paper is eventually

published in JGP, the files will be directly linked to specific figures in the published article.

Source Data Figures should be provided as individual PDF files (one file per figure). Authors should endeavor to retain a minimum resolution of 300 dpi or pixels per inch. Please review our instructions for export from Photoshop, Illustrator, and PowerPoint here: <https://rupress.org/jgp/pages/submission-guidelines#revised>

When revising your manuscript, please be sure it is a double-spaced MS Word file and that it includes editable tables, if appropriate.

Please submit your revised manuscript via this link:
<https://jgp.msubmit.net/cgi-bin/main.plex>

Thank you for the opportunity to consider your manuscript.

Sincerely,

David Eisner, D. Phil
On behalf of Journal of General Physiology

Journal of General Physiology's mission is to publish mechanistic and quantitative molecular and cellular physiology of the highest quality; to provide a best-in-class author experience; and to nurture future generations of independent researchers.

Reviewer #1 (Comments to the Authors):

This is a potentially interesting study that looks at possible homeostatic plasticity in the excitability of peripheral sensory neurons. The study uses both cultured mouse and cultured human sensory neurons to investigate how sustained depolarization alters intrinsic excitability. Interestingly, in both cases excitability is reduced. The decrease is fairly reversible. The mechanism is not entirely clear but seems to involve in part a reduction in sodium current density. The study also investigates if chronic (24 hr) stimulation using optogenetics can induce a similar alteration in excitability. Surprisingly the 1 Hz stimulation does not seem to have a significant impact. While the study is interesting, there are some areas that could be improved. One important issue is the physiological relevance of 30 mM KCl. The reduced excitability is characterized in the abstract as a compensatory decrease in excitability. However, 30 mM KCl does not induce increased action potentials, so it is not clear that this is really homeostatic plasticity or more of an injury response. This could be better addressed.

Related, while figure 1A inset shows the change in resting membrane potential after 5 minutes of 30 mM KCl, it is important to know what happens after 24 hours. Perhaps the cells compensate for the prolonged exposure and resting membrane potential partially recovers. It is important to know what the RMP (and spontaneous firing situation) is in the neurons at the 24 hr exposure period compared to control.

Another significant concern is the lack of consideration of a large body of literature indicating that injury and time in culture can impact excitability significantly. The process of culturing sensory neurons can have a significant impact on excitability. Please confirm if the neurons were cultured with NGF or other growth factors.

The data from the 1 Hz optogenetic stimulation suggests that chronic action potential activity does not induce homeostatic plasticity. However, the discussion suggests that this may be because the stimulation is artificial and therefore a more physiologically relevant stimulation is needed. This seems a biased interpretation. The simpler interpretation is that chronic action potential activity does not induce homeostatic scaling of excitability. On the other hand, it is not clear that the decrease in sodium current density observed with 24 hr of 30 mM KCl serves to sustain network function. Given that the chronic depolarization with KCl does not induce action potentials, the changes that occur with KCl may be related to elevated calcium levels rather than chronic action potentials, and may not represent homeostatic plasticity in the classic sense.

Minor issues:

Please clarify the age of the mice used for the DRG cultures.

Please clarify for figure 1A inset if none of the five cells exhibited any action potential activity in response to the 5 minute 30 mM KCl exposure.

Reviewer #2 (Comments to the Authors):

This manuscript from McIlvried and del Rosario et al. investigates whether sensory neurons from both mice and humans can undergo homeostatic plasticity. To accomplish this, they use a model of sustained depolarization whereby neurons are incubated in 30 mM KCl for 24 followed by a recovery period of 24h in culture media. The authors found no differences in viability induced by KCl incubation. Using current clamp electrophysiology, they quantified a variety of excitability parameters after a 24h recovery period in culture media. For mouse DRG neurons, excitability was noticeably reduced, as evidenced by a higher proportion of neurons firing only one action potential in response to step depolarizations, as well as changes in several other analyzed parameters that are indicative of reduced excitability. In human neurons, the effects were more subtle. The authors

also examined homeostatic plasticity using continual ChR2 stimulation for 24 h (1Hz), but found this protocol was not sufficient to induce any changes in excitability. To understand the mechanistic basis of the change in excitability induced by 30 mM KCl, the authors performed bulk RNAseq on cultured sensory neurons; however, no transcriptional changes were observed. Finally, the authors perform voltage-clamp recordings, and found a significant reduction in the whole-cell sodium current, but no change in potassium currents, following KCl incubation for 24h.

Overall, the paper is well written and easy to read, the figures are well made, the data are of high quality, and appropriate stats are used throughout. My main concern is the choice of model used (in vitro cultures, 30 mM KCl) and its physiological relevance, the latter of which the authors note is unclear. A new model should be developed that more accurately recapitulates putative physiological mechanisms of homeostatic plasticity in sensory neurons. This lab is well equipped to develop such models, which would be of great benefit to the scientific/sensory neuroscience community. However, in my opinion, the authors are appropriately careful with their interpretations.

The main thing lacking from this manuscript in its current state is mechanism. I strongly suggest the following experiments to bolster the authors' claims that changes in sodium channels are underlying the plasticity observed in mouse sensory neurons:

- The authors do not measure sodium currents 24 h following recovery. One would expect them to recover on average if they are driving the homeostatic change.
- The authors should include pharmacology to understand if the changes in sodium currents are driven primarily by TTX-resistant or TTX-sensitive sodium channels, or both.
- A more in depth sodium channel biophysical analysis could help shed light on whether certain sodium channel properties are altered following 24 h in KCl, or if this is simply an effect on current density. If that latter, a PanNav antibody could be used to quantify the total surface expression of Navs before, during, and after KCl treatment.

In addition to these experiments, the authors should show that DRG morphology is not affected by KCl treatment. They performed viability experiments which found no differences in survival following KCl treatment, but this does not preclude negative effects on cell health which could compromise data interpretation. Staining with a pan-neuronal marker, such as beta3-tubulin, is an easy way to quantitatively show morphology is not significantly changed by KCl incubation.

Reviewer #3 (Comments to the Authors):

The excitability of primary sensory neurons is critical for providing the appropriate input to the central nervous system to drive somatosensation and pain. It is known that changes in the excitability of a subset of nociceptive sensory neurons can be a contributing factor in pathological forms of pain. The excitation of primary sensory neurons (PSNs) normally originates in the peripheral terminals embedded in the tissues and is conducted along the axon. Under physiological conditions, action potentials are not initiated at the cell soma, but it is known that pathological conditions can lead to apparently spontaneous firing at the cell body. This suggests that, in certain circumstances, the excitability of PSNs can change due to intrinsic factors. Therefore, McIlvried et al. attempted to investigate the intrinsic homeostatic plasticity of PSNs. To this end, they characterized different types of cultured PSNs subjected to treatments that affect neuronal homeostasis.

First, they tested excitability changes in mouse PSNs after prolonged depolarization (30 mM KCl) and found an increased rheobase and a significant change in the distribution of firing patterns. These changes were reversed after 24 hours of culturing in standard conditions. Next, they tested homeostatic changes evoked by repeated stimulation (using optogenetic stimulation) and concluded that this treatment didn't significantly affect the excitability. Finally, the authors demonstrated that human sensory neurons undergo similar excitability changes as mouse PSNs. Bulk sequencing of cultured neurons didn't reveal any changes in gene expression following depolarization treatment. Finally, the authors demonstrated that the excitability change can be linked to a change in sodium current density.

The manuscript might be of interest to specialists interested in primary sensory neurons and, as such, merits publication. Of particular value are the recordings of cultured human sensory neurons, which are still relatively rare and highly informative. However, certain issues limit the manuscript's value from a physiologist's perspective. Addressing these issues would greatly improve the manuscript.

Major points

1. The authors characterize neurons in in vitro culture by evoking action potentials (APs) at the soma and measuring excitability changes there. It is unclear in what physiological situations the authors would expect primary sensory neurons (PSNs) to experience prolonged depolarizing conditions. For instance, would PSNs undergo such conditions at the peripheral terminal following local injury or at the cell body within the ganglia following major damage, like nerve transection? It would significantly help with the interpretation of the study to determine whether sensory neurons that experience pathological conditions in vivo (e.g., nerve transection) exhibit properties similar to neurons that were subjected to long-term depolarization conditions. What happens to the rheobase and the distribution of firing patterns in neurons isolated from such conditions? Do these neurons exhibit the same homeostatic changes, and do they recover their basal properties after exposure to KCl treatment, much like neurons isolated from healthy mice? Does local inflammation at the periphery (e.g. inflammatory model) induce homeostatic adjustment also in the cell body? Obtaining data on these points is crucial for attempting to relate the observed phenomena to physiological processes.

2. Primary sensory neurons (PSNs) are known to comprise a highly diverse population, as indicated by single-cell sequencing indicating distinct molecular characteristics among different nociceptor subpopulations. Therefore, it is crucial to ensure that experiments conducted under various paradigms target the same groups of neurons. In the current manuscript, different experiments employ varying approaches for the selection of PSN subpopulations. Initially, neurons are chosen based on their size (Figures 1-3), followed by a limitation to genetically defined Trpv1 lineage neurons (Figure 4). For sequencing, the entire population is sampled (Figure 6). This diversity in selection methods is evidently affecting the data, as seen in, for example, the substantial difference in the distribution of firing patterns between the control population in Figure 3D and that in Figure 4P. To address this issue, it may be beneficial to strive for more consistent controls across different approaches. This could involve either conducting optogenetic experiments with a panneuronal labeling line (e.g., Adv:Cre) or demonstrating that neurons within the Trpv1:Cre lineage exhibit a similar response in KCl experiments to neurons selected from the total population.

Minor points

1. On p14 authors state 'The neurons that did not fire any AP up to the maximum injected current ramp of 1nA were classified as non-responders (NR; Fig. 1B-right and 1D-top) and their rheobase was set to 1nA for analysis (Fig. 1D top of the graph).' I am not convinced that this approach is statistically sound, authors should either justify it better or compare fraction of responding versus non-responding neurons and then rheobase value of responding neurons. Similar approach is used in rheobase measurements during human sensory neurons recordings (Fig 5E).

2. I think that since PSNs neurons lack significant synaptic input extensive discussion of synaptic scaling seems not relevant to questions handled by the manuscript.

Dear Dr. Eisner,

We thank the reviewers for carefully evaluating our manuscript. Below you will find our responses to the reviewers' critiques in blue typeset. Additionally, in the revised manuscript, we labeled substantial changes in response to reviewers' comments with red typeset. We believe our revised manuscript has improved substantially, and we sincerely hope the reviewers will find our work acceptable for publication in the Journal of General Physiology.

Sincerely,

Robert Gereau

General Revisions:

Thank you for submitting your manuscript, entitled "Intrinsic Homeostatic Plasticity in Mouse and Human Sensory Neurons" to JGP. Your manuscript has now been seen by 3 reviewers, whose comments are appended below.

We sincerely thank all the reviewers for the time and consideration they took to review this manuscript.

You will see that the reviewers have raised several concerns that should be addressed prior to further consideration of the manuscript at JGP. In particular, I would identify the following. (1) The differences between the results of optogenetic stimulation and 30 mM KCl. The ms suggests that the former may not represent physiological activity. While this may be the case, the reviewers mention that 30 mM KCl is certainly not a simple model of increased electrical activity. At the very least, this point should be discussed.

We agree that 30 mM KCl is not a simple model of increased electrical activity, rather it is a tool and classic approach to drive continuous depolarization in neurons to study homeostatic plasticity. We did not try to mimic physiological or pathological pain conditions in culture but rather understand how sensory neurons respond to abrupt changes in depolarization. This model has been extensively used by many researchers studying mechanisms of homeostatic plasticity in the central nervous system (CNS). In our manuscript, we have cited their seminal work and discussed whether primary sensory neurons in the peripheral nervous system (PNS) are also capable of intrinsically adjusting their excitability in conditions of long-lasting depolarization (30mM KCl for 24hr). Variations of this concern were brought up by all 3 reviewers, so we have reemphasized important points in the manuscript (see *introduction and discussion sections*) to clarify the issue. See our more in-depth, detailed responses to the individual comments.

Furthermore, it seems essential to know what effect 30 mM KCl would have in the TRPV1 neurons used for the optogenetic experiments.

We agree with the reviewers and have performed additional experiments to address this directly. Our data show that the majority of TRPV1-lineage neurons in control media fired an action potential (AP) to ramp depolarization (9/11 or 81%), while significantly fewer neurons fired APs to ramp depolarization after 24h incubation in 30mM KCl (2/12 or 17%), **p=0.0033 fisher's exact test. In addition, KCl-treated neurons had a significant increase in rheobase compared to neurons in control

media in response to ramp depolarizations (** $p=0.0052$, Mann-Whitney test). We also found an increase in rheobase in KCl-treated neurons compared to neurons in control media subjected to step current injections ($p=0.0177$, Mann-Whitney test). These findings are very similar to our initial data on small-diameter mouse DRG neurons (*these data have been added to the results section; see lines 442-452*). See our further detailed response below to the individual reviewer's comment.

As pointed out by reviewer #2, there is a need to show reversibility of the effect of 30 mM KCl on the sodium current and that reviewer makes other useful suggestions to probe mechanism.

We agree with the reviewer and have performed these experiments. Our data show that sodium currents, in neurons previously treated with 30mM KCl for 24h, were fully recovered once they were placed back in fresh control media for an additional 24 hr (*Fig. 6B*). These data have been added to the results section; see lines 530-531.

Reviewer #1 (Comments to the Authors):

This is a potentially interesting study that looks at possible homeostatic plasticity in the excitability of peripheral sensory neurons. The study uses both cultured mouse and cultured human sensory neurons to investigate how sustained depolarization alters intrinsic excitability. Interestingly, in both cases excitability is reduced. The decrease is fairly reversible. The mechanism is not entirely clear but seems to involve in part a reduction in sodium current density. The study also investigates if chronic (24 hr) stimulation using optogenetics can induce a similar alteration in excitability. Surprisingly the 1 Hz stimulation does not seem to have a significant impact. While the study is interesting, there are some areas that could be improved.

One important issue is the physiological relevance of 30 mM KCl. The reduced excitability is characterized in the abstract as a compensatory decrease in excitability. However, 30 mM KCl does not induce increased action potentials, so it is not clear that this is really homeostatic plasticity or more of an injury response. This could be better addressed.

We thank the reviewer for bringing up this important point. We apologize that this point was not more clear in the submitted manuscript. To clarify, the point of our study here was to assess whether homeostatic plasticity is present in mouse and human sensory neurons. To do this, we utilize 30mM KCl-induced depolarization for 24h as a tool to probe these mechanisms. This depolarization protocol is the standard one used in the homeostatic plasticity field, and thus we saw this as a useful starting point for these studies. KCl treatment in vitro is a classic approach to studying homeostatic plasticity in the CNS, extensively documented by the laboratories of Eve Marder, Gina Turrigano and others. It is used as a tool to “push” neurons, see if they adjust to the stimulus, and re-adjust back after the stimulus has been removed (thus demonstrating homeostatic plasticity). These studies have shown that KCl drives long-lasting depolarization and intrinsic changes in the excitability of various CNS neurons (Leslie et al., 2001; O'Leary et al., 2010; He et al., 2020). Homeostatic regulation of intrinsic excitability can control neuronal activity by dynamically modulating their intrinsic excitability through voltage-gated ion channel expression and function. The effects described in this manuscript fit the classic definition of intrinsic homeostatic plasticity, wherein a sustained stimulus induces compensatory changes in excitability, which importantly recovers after the stimulus is removed. While it is true that the culturing process can induce injury to the neurons, the control groups were also cultured neurons from the same preps as the KCl-treated neurons, so the effects observed were due to treatment and not an injury response.

Additionally, it is important to note that in our study we did not try to mimic physiological conditions of pain. High concentrations of KCl in vitro is not a model of pain, but, as mentioned above, it was used as a tool to drive a sustained depolarizing stimulus to investigate if sensory neurons have the ability to intrinsically counteract changes in excitability similar to CNS neurons. If so, in future investigations this could allow us to study the underlying mechanisms of adaptive plasticity in mouse and human DRG neurons and their potential dysregulation during a maladaptive hyperexcitable neuronal state as reported in sensory neurons associated with chronic pain conditions.

We have emphasized these important points and changed some of the language to increase clarity in the manuscript (see lines 99-106 and 561-565).

Related, while figure 1A inset shows the change in resting membrane potential after 5 minutes of 30 mM KCl, it is important to know what happens after 24 hours. Perhaps the cells compensate for the prolonged exposure and resting membrane potential partially recovers. It is important to know what the RMP (and spontaneous firing situation) is in the neurons at the 24 hr exposure period compared to control.

This is an excellent point, and we have performed additional experiments to address this. We found that sensory neurons exposed to 30mM KCl for 24hr maintained a depolarized membrane potential of -18 ± 1.3 mV. This membrane potential quickly recovered after cells are returned to the control extracellular solution to about -48 ± 1.7 mV ($n=11$, $****p<0.0001$). *This has been indicated in the manuscript in the result section corresponding to Fig.1; see lines 312-319.*

Another significant concern is the lack of consideration of a large body of literature indicating that injury and time in culture can impact excitability significantly. The process of culturing sensory neurons can have a significant impact on excitability. Please confirm if the neurons were cultured with NGF or other growth factors.

We agree that injury and time in culture can cause changes in the excitability of sensory neurons. For that reason, in our studies, we included appropriate controls of neurons in culture for the same period as the neurons treated with KCl to maintain similar and comparative recording conditions. Furthermore, neurons in both control and treatment groups were from the same culture preparations to account for variability in culturing between preps. *This has been clarified in the methods section (see lines 173-178).*

The sensory neuron cultures were cultured in DRG media without NGF (*this is now noted in the method section, see lines 147, 159*). Although we do not see substantial differences in mouse and human DRG neurons' membrane properties associated with days in vitro, we have also cited appropriate literature describing this phenomenon (Klein et al., 1991; Delrée et al., 1993; Gold et al., 1996; Black et al., 1999; Davidson et al., 2014; Black et al., 2018) (*method section, see lines 174-175*).

The data from the 1 Hz optogenetic stimulation suggests that chronic action potential activity does not induce homeostatic plasticity. However, the discussion suggests that this may be because the stimulation is artificial and therefore a more physiologically relevant stimulation is needed. This seems a biased interpretation. The simpler interpretation is that chronic action potential activity does not induce homeostatic scaling of excitability.

We agree with the reviewer and have modified this statement accordingly (see lines 607-609).

On the other hand, it is not clear that the decrease in sodium current density observed with 24 hr of 30 mM KCl serves to sustain network function.

In Fig.6B, we have added new data showing that sodium current density completely recovers after KCl-treated cells are replaced with fresh control media for 24hr. Given that the changes in excitability of sensory neurons follows the same trajectory when treated with high KCl, we suggest that sodium channels may play an essential role in modulating sensory neuron excitability after sustained depolarization.

Given that the chronic depolarization with KCl does not induce action potentials, the changes that occur with KCl may be related to elevated calcium levels rather than chronic action potentials, and may not represent homeostatic plasticity in the classic sense.

The reviewer is correct; it is well established that a 30mM KCl stimulus evokes an increase in intracellular calcium. Elevated calcium levels during chronic depolarization with KCl may be important for the compensatory decrease in excitability, which we have discussed in the Discussion section of the revised manuscript. However, it is still an intrinsic homeostatic response of the neuron as a result of chronic depolarization, regardless of increased calcium levels that occur downstream of the KCl stimulus. The effects described in this manuscript fit the classic definition of intrinsic homeostatic plasticity—a sustained stimulus for 24h that induces a compensatory change in excitability, which importantly recovers after the stimulus is removed.

Minor issues:

Please clarify the age of the mice used for the DRG cultures.

The age of the mice used for the DRG cultures is clarified in the method section of the revised manuscript under “Animals.”

Please clarify for figure 1A inset if none of the five cells exhibited any action potential activity in response to the 5 minute 30 mM KCl exposure.

None of the cells treated with 30mM KCl for 5 mins exhibited any action potential activity. This has been clarified in the result section for Fig.1 (see *lines 315-317*)

Reviewer #2 (Comments to the Authors):

This manuscript from McIlvried and del Rosario et al. investigates whether sensory neurons from both mice and humans can undergo homeostatic plasticity. To accomplish this, they use a model of sustained depolarization whereby neurons are incubated in 30 mM KCl for 24 followed by a recovery period of 24h in culture media. The authors found no differences in viability induced by KCl incubation. Using current clamp electrophysiology, they quantified a variety of excitability parameters after a 24h recovery period in culture media. For mouse DRG neurons, excitability was noticeably reduced, as evidenced by a higher proportion of neurons firing only one action potential in response to step depolarizations, as well as changes in several other analyzed parameters that are indicative of reduced excitability. In human neurons, the effects were more subtle. The authors also examined homeostatic plasticity using continual ChR2 stimulation for 24 h (1Hz), but found this protocol was not sufficient to induce any changes in excitability. To understand the mechanistic basis of the change in excitability induced by 30 mM KCl, the authors performed bulk RNAseq on cultured sensory neurons;

however, no transcriptional changes were observed. Finally, the authors perform voltage-clamp recordings, and found a significant reduction in the whole-cell sodium current, but no change in potassium currents, following KCl incubation for 24h.

Overall, the paper is well written and easy to read, the figures are well made, the data are of high quality, and appropriate stats are used throughout.

We thank the reviewer for recognizing the extensive work that was put into this manuscript, and for their thoughtful suggestions. We have addressed the reviewer's concerns below and adjusted the manuscript accordingly.

My main concern is the choice of model used (in vitro cultures, 30 mM KCl) and its physiological relevance, the latter of which the authors note is unclear.

We thank the reviewer for bringing up this important point. We apologize for any confusion, we are not arguing that 30mM KCl for 24h is physiologically relevant- it is just a stimulus (the standard one in the homeostatic plasticity field). KCl treatment in vitro is a classic approach to studying homeostatic plasticity in the CNS, extensively documented by the laboratories of Eve Marder, Gina Turrigano and others. It is used as a tool to “push” neurons, see if they adjust to the stimulus, and re-adjust back after the stimulus has been removed (thus demonstrating homeostatic plasticity). These prior studies have shown that KCl drives long-lasting depolarization and intrinsic changes in the excitability of neurons in the CNS (Leslie et al., 2001; O'Leary et al., 2010; He et al., 2020). Homeostatic regulation of intrinsic excitability can dynamically modulate intrinsic excitability through alterations in voltage-gated ion channel expression and function. The effects described in this manuscript represent an important first step and fit the classic definition of intrinsic homeostatic plasticity—a sustained stimulus that induces a compensatory change in excitability, which importantly recovers after the stimulus is removed.

Additionally, it is important to note that in our study we did not try to mimic physiological conditions of pain. Exposure to high concentrations of KCl in vitro does not represent a model of pain, but, as mentioned above, it was used as a tool to drive a sustained depolarizing stimulus to investigate if sensory neurons have the ability to intrinsically counteract changes in excitability similar to CNS neurons. If so, in future investigations this could allow us to study the underlying mechanisms of adaptive plasticity in mouse and human DRG neurons and their potential dysregulation during a maladaptive hyperexcitable neuronal state as reported in sensory neurons associated with chronic pain conditions.

A new model should be developed that more accurately recapitulates putative physiological mechanisms of homeostatic plasticity in sensory neurons. This lab is well equipped to develop such models, which would be of great benefit to the scientific/sensory neuroscience community. However, in my opinion, the authors are appropriately careful with their interpretations.

We agree with the reviewer that it will be important to develop a new model/strategy to help evaluate mechanisms of homeostatic plasticity in response to acute and chronic pain conditions. However, we suggest that this important question is beyond the scope of this initial paper. This will certainly be an interesting line of research to pursue in the future and we have added a comment on this point to the discussion (*see lines 700-709*).

The main thing lacking from this manuscript in its current state is mechanism. I strongly suggest the following experiments to bolster the authors' claims that changes in sodium channels are underlying the plasticity observed in mouse sensory neurons:

- The authors do not measure sodium currents 24 h following recovery. One would expect them to recover on average if they are driving the homeostatic change.

An excellent point - We agree with the reviewer and have performed additional experiments to assess this directly. Our data show that sodium currents, in cells previously treated with 30mM KCl for 24h, were fully recovered once they were placed back in fresh control media for an additional 24 hr (Fig. 6B).

- The authors should include pharmacology to understand if the changes in sodium currents are driven primarily by TTX-resistant or TTX-sensitive sodium channels, or both.
- A more in depth sodium channel biophysical analysis could help shed light on whether certain sodium channel properties are altered following 24 h in KCl, or if this is simply an effect on current density. If that latter, a PanNav antibody could be used to quantify the total surface expression of Navs before, during, and after KCl treatment.

We agree with the reviewer that these are interesting and important questions for further study. We feel that the extensive body of information provided here represents an important first step, and we believe that a full pharmacological and biophysical characterization of sodium channels is outside the scope of this manuscript. We intend to follow up on these precise questions in future studies, but there will be extensive work to characterize these changes, suitable for a follow-up manuscript.

In addition to these experiments, the authors should show that DRG morphology is not affected by KCl treatment. They performed viability experiments which found no differences in survival following KCl treatment, but this does not preclude negative effects on cell health which could compromise data interpretation. Staining with a pan-neuronal marker, such as beta3-tubulin, is an easy way to quantitatively show morphology is not significantly changed by KCl incubation.

We thank the reviewer for this recommendation. In addition to our cell viability assay, our data show there were no statistical differences in RMP between control media-treated and KCl-treated cells (Table 1 and Table 2). To further address the recommendation of the reviewer, we performed additional immunocytochemistry studies using beta3-tubulin as a marker to assess any morphological changes induced by KCl treatment. We find no statistical differences in the distribution of DRG neuron size between control media and KCl-treated neurons (Fig. S1A-C).

Reviewer #3 (Comments to the Authors):

The excitability of primary sensory neurons is critical for providing the appropriate input to the central nervous system to drive somatosensation and pain. It is known that changes in the excitability of a subset of nociceptive sensory neurons can be a contributing factor in pathological forms of pain. The excitation of primary sensory neurons (PSNs) normally originates in the peripheral terminals embedded in the tissues and is conducted along the axon. Under physiological conditions, action potentials are not initiated at the cell soma, but it is known that pathological conditions can lead to apparently spontaneous firing at the cell body. This suggests that, in certain circumstances, the excitability of PSNs can change due to intrinsic factors. Therefore, McIlvried et al. attempted to investigate the intrinsic homeostatic plasticity of PSNs. To this end, they characterized different types of cultured PSNs subjected to treatments that affect neuronal homeostasis.

First, they tested excitability changes in mouse PSNs after prolonged depolarization (30 mM KCl) and found an increased rheobase and a significant change in the distribution of firing patterns. These changes were reversed after 24 hours of culturing in standard conditions. Next, they tested homeostatic changes evoked by repeated stimulation (using optogenetic stimulation) and concluded that this treatment didn't significantly affect the excitability. Finally, the authors demonstrated that human sensory neurons undergo similar excitability changes as mouse PSNs. Bulk sequencing of cultured neurons didn't reveal any changes in gene expression following depolarization treatment. Finally, the authors demonstrated that the excitability change can be linked to a change in sodium current density.

The manuscript might be of interest to specialists interested in primary sensory neurons and, as such, merits publication. Of particular value are the recordings of cultured human sensory neurons, which are still relatively rare and highly informative.

We thank the reviewer for acknowledging the significance of our data, and for their perceptive comments. We have addressed the reviewer's concerns below and adjusted the manuscript accordingly.

However, certain issues limit the manuscript's value from a physiologist's perspective. Addressing these issues would greatly improve the manuscript.

Major points

1. The authors characterize neurons in in vitro culture by evoking action potentials (APs) at the soma and measuring excitability changes there. It is unclear in what physiological situations the authors would expect primary sensory neurons (PSNs) to experience prolonged depolarizing conditions. For instance, would PSNs undergo such conditions at the peripheral terminal following local injury or at the cell body within the ganglia following major damage, like nerve transection? It would significantly help with the interpretation of the study to determine whether sensory neurons that experience pathological conditions in vivo (e.g., nerve transection) exhibit properties similar to neurons that were subjected to long-term depolarization conditions. What happens to the rheobase and the distribution of firing patterns in neurons isolated from such conditions? Do these neurons exhibit the same homeostatic changes, and do they recover their basal properties after exposure to KCl treatment, much like neurons isolated from healthy mice? Does local inflammation at the periphery (e.g. inflammatory model) induce homeostatic adjustment also in the cell body? Obtaining data on these points is crucial for attempting to relate the observed phenomena to physiological processes.

This reviewer raises important questions here, really identifying a large scope of work that we hope to address as a follow-up to the findings we hope to report in this initial manuscript. In the response to reviewers 1 and 2 above, we have discussed the rationale for the use of the 30mM KCl stimulus as a tool to assess whether sensory neurons of the DRG undergo homeostatic plasticity, so we will not repeat those discussions here. The other points regarding the potential for injury or inflammation to induce similar adaptive responses are critically important, and form the basis of substantial efforts in planned future investigation. We would like to suggest that such studies are beyond the scope of the present manuscript, which will lay the groundwork and demonstrates that indeed sensory neurons from mice and humans engage homeostatic mechanisms to regulate intrinsic excitability. As we have mentioned in the discussion of the revised manuscript – it will be very interesting to learn whether

these mechanisms are engaged or disrupted in the context of pathologies leading to conditions such as chronic pain and itch. With these foundational findings in hand, future investigations will address the underlying mechanisms of adaptive homeostatic plasticity in mouse and human DRG neurons and their potential dysregulation during maladaptive hyperexcitable neuronal states as reported in sensory neurons associated with chronic pain conditions.

2. Primary sensory neurons (PSNs) are known to comprise a highly diverse population, as indicated by single-cell sequencing indicating distinct molecular characteristics among different nociceptor subpopulations. Therefore, it is crucial to ensure that experiments conducted under various paradigms target the same groups of neurons. In the current manuscript, different experiments employ varying approaches for the selection of PSN subpopulations. Initially, neurons are chosen based on their size (Figures 1-3), followed by a limitation to genetically defined Trpv1 lineage neurons (Figure 4). For sequencing, the entire population is sampled (Figure 6). This diversity in selection methods is evidently affecting the data, as seen in, for example, the substantial difference in the distribution of firing patterns between the control population in Figure 3D and that in Figure 4P. To address this issue, it may be beneficial to strive for more consistent controls across different approaches. This could involve either conducting optogenetic experiments with a panneuronal labeling line (e.g., Adv:Cre) or demonstrating that neurons within the Trpv1:Cre lineage exhibit a similar response in KCl experiments to neurons selected from the total population.

Excellent suggestion. The TRPV1:Cre lineage mouse line targets the majority of neurons in the DRG (75-80%), and the majority of nociceptive neurons. As we were also still selecting for small diameter size in these neurons as we did for the studies of neurons from wild-type mice, we believe that the majority of sensory neurons that we patched in Figure 1 were also part of the TRPV1: Cre lineage. However, to address this concern directly, we conducted additional experiments to test whether Trpv1:Cre lineage neurons exhibit a similar response to KCl. Our data show that the majority of TRPV1 lineage neurons in control media fired an action potential (AP) to ramp depolarization (9/11 or 81%), while significantly fewer neurons fired APs to ramp depolarization after 24h incubation in 30mM KCl (2/12 or 17%), $**p=0.0033$ fisher's exact test. In addition, KCl-treated neurons had a significant increase in rheobase compared to neurons in control media in response to ramp depolarizations ($p=0.0052$). We also found an increase in rheobase in KCl-treated neurons compared to neurons in control media subjected to step current injections ($*p=0.0177$). These findings are very similar to our initial data on small-diameter mouse DRG neurons from wild-type mice (*we have included these data in the results section; see lines 442-452*). We therefore suggest that both sets of experiments are likely targeting similar populations of neurons, and that small diameter mouse DRG neurons showed significant inhibition of intrinsic excitability following prolonged depolarization, but not action potential firing.

Minor points

1. On p14 authors state 'The neurons that did not fire any AP up to the maximum injected current ramp of 1nA were classified as non-responders (NR; Fig. 1B-right and 1D-top) and their rheobase was set to 1nA for analysis (Fig. 1D top of the graph).' I am not convinced that this approach is statistically sound, authors should either justify it better or compare fraction of responding versus non-responding neurons and then rheobase value of responding neurons.

The fraction of responding versus non-responding neurons is located in Fig.1C. While we used the strategy of assigning non-responders (NRs) to the max for our rheobase-ramp protocol so that we weren't excluding an important and large data set of NRs into the analysis, there also was a

significant change in rheobase when considering only the population of neurons that do fire action potentials in response to ramp depolarization (responders) from control media compared to high KCl media (Fig.S1D, $**p=0.0032$).

Similar approach is used in rheobase measurements during human sensory neurons recordings (Fig 5E).

Similarly, there was a trend toward an increase in rheobase from control media compared to high KCl media when the NRs were removed from the analysis, though this did not reach statistical significance (Fig.S2E, $p=0.0558$) for the human neurons.

2. I think that since PSNs neurons lack significant synaptic input extensive discussion of synaptic scaling seems not relevant to questions handled by the manuscript.

We use the detailed discussion of synaptic scaling as a preamble to introduce the overall picture of the manuscript and the significance of the research in other types of cells. If the editors prefer, we can remove this discussion from the manuscript.

REFERENCES

- Black, B.J., R. Atmaramani, R. Kumaraju, S. Plagens, M. Romero-Ortega, G. Dussor, T.J. Price, Z.T. Campbell, and J.J. Pancrazio. 2018. Adult mouse sensory neurons on microelectrode arrays exhibit increased spontaneous and stimulus-evoked activity in the presence of interleukin-6. *J Neurophysiol.* 120:1374-1385.
- Black, J.A., T.R. Cummins, C. Plumpton, Y.H. Chen, W. Hormuzdiar, J.J. Clare, and S.G. Waxman. 1999. Upregulation of a silent sodium channel after peripheral, but not central, nerve injury in DRG neurons. *J Neurophysiol.* 82:2776-2785.
- Davidson, S., B.A. Copits, J. Zhang, G. Page, A. Ghetti, and R.W.t. Gereau. 2014. Human sensory neurons: Membrane properties and sensitization by inflammatory mediators. *Pain.* 155:1861-1870.
- Delrée, P., C. Ribbens, D. Martin, B. Rogister, P.P. Lefebvre, J.M. Rigo, P. Leprince, J. Schoenen, and G. Moonen. 1993. Plasticity of developing and adult dorsal root ganglion neurons as revealed in vitro. *Brain Res Bull.* 30:231-237.
- Gold, M.S., S. Dastmalchi, and J.D. Levine. 1996. Co-expression of nociceptor properties in dorsal root ganglion neurons from the adult rat in vitro. *Neuroscience.* 71:265-275.
- He, L.S., M.C.P. Rue, E.O. Morozova, D.J. Powell, E.J. James, M. Kar, and E. Marder. 2020. Rapid adaptation to elevated extracellular potassium in the pyloric circuit of the crab, *Cancer borealis*. *J Neurophysiol.* 123:2075-2089.
- Klein, C.M., O. Guillaumondegui, C.D. Krennek, R.A. La Forte, and R.E. Coggeshall. 1991. Do neuropeptides in the dorsal horn change if the dorsal root ganglion cell death that normally accompanies peripheral nerve transection is prevented? *Brain Res.* 552:273-282.
- Leslie, K.R., S.B. Nelson, and G.G. Turrigiano. 2001. Postsynaptic depolarization scales quantal amplitude in cortical pyramidal neurons. *The Journal of neuroscience : the official journal of the Society for Neuroscience.* 21:Rc170.
- O'Leary, T., M.C. van Rossum, and D.J. Wyllie. 2010. Homeostasis of intrinsic excitability in hippocampal neurones: dynamics and mechanism of the response to chronic depolarization. *The Journal of physiology.* 588:157-170.

July 3, 2024

Dr. Robert W Gereau IV
Washington University in St. Louis School of Medicine
Anesthesiology
660 S Euclid Ave
Campus Box 8054
St Louis, MO 63110

Re: 202313488R1

Dear Dr. Gereau IV,

Thank you for submitting your manuscript, entitled "Intrinsic Homeostatic Plasticity in Mouse and Human Sensory Neurons" to JGP. Your manuscript has now been seen by 3 reviewers, whose comments are appended below. Reviewer #1 reinforces his/her comment on the previous version that there is no evidence that the phenomenon you are investigating is actually "homeostatic plasticity". The editors have discussed this and agree. This will require modification of the text, including the title. Please also note the other comments below.

We would be pleased to receive a suitably revised manuscript that addresses these concerns, which will be re-reviewed, most likely by some or all of the original referees. In addition, please do not hesitate to contact me (via the editorial office) if you feel that a discussion of the reviewers' and editors' comments would be helpful.

Please submit your revised manuscript via the link below along with a point-by-point letter that details your responses to the editors' and reviewers' comments, as well as a copy of the text with alterations highlighted (boldfaced or underlined). If the article is eventually accepted, it would include a 'revised date' as well as submitted and accepted dates. If we do not receive the revised manuscript within one year, we will regard the article as having been withdrawn. We would be willing to receive a revision of the manuscript at a later time, but the manuscript will then be treated as a new submission, with a new manuscript number.

Please pay particular attention to recent changes to our instructions to authors in the following sections: Data presentation, Blinding and randomization and Statistical analysis, under Materials and Methods, as shown here: <https://rupress.org/jgp/pages/submission-guidelines#prepare>. Re-review will be contingent on inclusion of the required information (including for data added during revision) and demonstration of the experimental reproducibility of the results. Also, to improve the reproducibility of published content, we have partnered with SciScore. Authors are prompted in eJP to copy and paste the Materials and Methods section of their manuscript for a SciScore assessment when submitting their revised manuscript. Authors are encouraged (not required) to further revise their Materials and Methods if the SciScore is below 4. More information can be found here: <https://rupress.org/jgp/pages/submission-guidelines#sciscore>

Please note, JGP now requires authors to submit Source Data used to generate figures containing gels and Western blots with all revised manuscripts (when applicable). This Source Data consists of fully uncropped and unprocessed images for each gel/blot displayed in the main and supplemental figures. If your paper includes cropped gel and/or blot images, please be sure to provide one Source Data file for each figure that contains gels and/or blots along with your revised manuscript files. File names for Source Data figures should be alphanumeric without any spaces or special characters (i.e., SourceDataF#, where F# refers to the associated main figure number or SourceDataFS# for those associated with Supplementary figures). The lanes of the gels/blots should be labeled as they are in the associated figure, the place where cropping was applied should be marked (with a box), and molecular weight/size standards should be labeled wherever possible. Source Data files will be made available to reviewers during evaluation of revised manuscripts and, if your paper is eventually published in JGP, the files will be directly linked to specific figures in the published article.

Source Data Figures should be provided as individual PDF files (one file per figure). Authors should endeavor to retain a minimum resolution of 300 dpi or pixels per inch. Please review our instructions for export from Photoshop, Illustrator, and PowerPoint here: <https://rupress.org/jgp/pages/submission-guidelines#revised>

When revising your manuscript, please be sure it is a double-spaced MS Word file and that it includes editable tables, if appropriate.

Please submit your revised manuscript via this link:
Link Not Available

Thank you for the opportunity to consider your manuscript.

Sincerely,

David Eisner, D. Phil
On behalf of Journal of General Physiology

Journal of General Physiology's mission is to publish mechanistic and quantitative molecular and cellular physiology of the highest quality; to provide a best-in-class author experience; and to nurture future generations of independent researchers.

Reviewer #1 (Comments to the Authors):

The revised manuscript has addressed several of the prior concerns, but parts of the response to the reviewers are not convincing. In particular, there are still concerns about the reliance on 30 mM KCl to explore adaptations in neuronal excitability. The authors indicate that 30 mM KCl was used to explore whether homeostatic plasticity might occur and provide historical context for its use. However, the problem with the current study is encapsulated in the concluding line of the abstract: "The finding that sensory neurons in mouse and human can undergo this type of homeostatic plasticity raises interesting questions regarding the potential role of this plasticity in normalization of sensory function in the context of prolonged activation of sensory neurons such as can occur in the context of chronic pain and itch."

1) A key question is whether or not the changes observed with chronic depolarization represent homeostatic plasticity. Homeostatic plasticity is often defined as alterations in intrinsic excitability through regulation of ion channel properties that normalize activity following perturbation of activity. 30 mM potassium depolarizes the membrane potential. If activity was normalized by some form of homeostatic plasticity, one might expect that the membrane potential would recover in face of the 30 mM KCl stimulus. That is not the case. A reduction in sodium current density is observed, but that does not result in a normalization of sensory function. Some form of plasticity is observed, but it does not seem to be homeostatic plasticity.

2) The second part of the final sentence of the abstract seems to suggest that the adaptation observed might be relevant to adaptations that might "occur in the context of chronic pain and itch." However, the response to the reviewers clearly indicates that 30 mM KCl is not being used to mimic physiological conditions of pain. Based on this, the relevance of the findings are unclear and the main conclusion of the study is unjustified.

The study concludes that the changes observed are adaptive and represent homeostatic plasticity. Given that chronic 30 mM KCl did not produce a compensatory change in the regulation of membrane potential, a change that allows recovery of normal excitability, a reasonable conclusion might be that sensory neurons do not undergo homeostatic plasticity. Indeed, given that sustained stimulation of action potentials did not produce any compensatory changes in intrinsic excitability, it appears that sensory neurons do not adapt to maintain a basal level of excitability. The study does not explicitly indicate what aspect of excitability is being normalized by the reduction in sodium currents. Indeed, the chronically depolarized neurons could be considered unexcitable and the reduction in sodium currents only furthers the reduced excitability. Thus the conclusion that homeostatic plasticity occurred is not justified. The assertion that homeostatic plasticity occurred raises the question of whether or not any change in any ion channel property would be construed as evidence of homeostatic plasticity.

Minor comment:

DIV (page 7, line 163) should be defined where it is first used.

Reviewer #2 (Comments to the Authors):

No further concerns noted. Very nice paper!

Reviewer #3 (Comments to the Authors):

The reviews have significantly improved the manuscript, and my comments have been satisfactorily addressed. One minor comment: I appreciate additional recordings of Trpv1:ChR2 neurons following prolonged depolarization. However, numerical data for this experiment were omitted from the manuscript, and only p values are given in the text without figures' references. This is probably an oversight and should be corrected.

Thank you for the reviews of our revised manuscript. The remaining major issue raised by reviewer 1 is a new perspective that we agree deserves attention in our manuscript.

The type of plasticity we report here for mouse and human sensory neurons is very similar to homeostatic plasticity mechanisms that have been previously reported by other groups for neurons in the CNS. We discuss this in detail in the manuscript, but after reading the reviewer's comment, we understand the viewpoint. Accordingly, we now refer to the changes we observe as "adaptive plasticity." We have made changes to the title, abstract, and throughout the manuscript, and have expanded the discussion to highlight this distinction (Lines 579-597).

To summarize - we understand the reviewer's perspective. We appreciate the more nuanced interpretation that the reviewer proposes and agree that this is an important issue to address. We propose the following changes to our manuscript that address the points made by the reviewer and should generate productive discussion in the field:

1. We changed the title to read as: "Intrinsic **Adaptive** Plasticity of Mouse and Human Sensory Neurons." (Line 1 in red typeset)
2. We updated the abstract to reflect this change.
3. We added a section to the discussion that addresses the reviewer's perspective regarding the lack of homeostasis with respect to the inciting stimulus – depolarization (Lines 579-597 and 641-650 in red typeset).

We thank the reviewer for this perspective on the data – and indeed on the field – related to homeostatic regulation. This is a point worth making and we are happy to include this in our manuscript.

September 24, 2024

Dr. Robert W Gereau IV
Washington University in St. Louis School of Medicine
Anesthesiology
660 S Euclid Ave
Campus Box 8054
St Louis, MO 63110

Re: 202313488R2

Dear Dr. Gereau IV,

I am pleased to let you know that your manuscript, entitled "Intrinsic Adaptive Plasticity in Mouse and Human Sensory Neurons" is scientifically acceptable for publication in Journal of General Physiology. Formal acceptance will follow when it is modified in accordance with the referees' remarks and our editorial policies.

Please note items that need attention are listed at the bottom of this email (under 'manuscript formatting checklist') and on the attached marked-up pdf file. Please also be sure to include a letter addressing the reviewers' comments point-by-point (if applicable) and a copy of the text with alterations highlighted (boldfaced or underlined). Your manuscript should be a double-spaced MS Word file and include editable tables, if appropriate.

Lastly, JGP requires a data availability statement for all research article submissions. These statements will be published in the article directly above the Acknowledgments. The statement should address all data underlying the research presented in the manuscript. Please visit the JGP instructions for authors for guidelines and examples of statements at <https://rupress.org/jgp/pages/editorial-policies#data-availability-statement>.

Please submit your final files via this link:
Link Not Available

Thank you for choosing to publish your research in JGP and please feel free to contact me with any questions.

Sincerely,

David Eisner, D. Phil
On behalf of Journal of General Physiology

Journal of General Physiology's mission is to publish mechanistic and quantitative molecular and cellular physiology of the highest quality; to provide a best in class author experience; and to nurture future generations of independent researchers.

Manuscript formatting checklist:

- MS Word document of text needed (including editable tables)
- MS Word document of supplemental text needed, if applicable (including figure legends and editable tables)
- Brief Statement describing supplementary information needed, if applicable (in subsection at end of Materials & Methods)
- Please include a data availability statement preceding the Acknowledgments section. Please see <https://rupress.org/jgp/pages/editorial-policies#data-availability-statement>
- Figures created at sufficient resolution and in acceptable format (including supplemental if applicable). If working in Illustrator, we prefer .ai or .eps file format. If working in Photoshop please use 600dpi/1000dpi .tiff or .psd file format. Minimum resolution at estimated print size: Minimum resolution for all figures is 600 dpi. For figures that contain both photographs and line art or text, 600 dpi is highly recommended. Figures containing only black and white elements (line art, no color, and no gray) should be 1,000 dpi. Maximum figure size is 7 in wide x 9 in high (17.5 x 22.8 cm) at the correct resolution. <https://jgp.rupress.org/fig-vid-guidelines>
- Supplemental figures, if any, conforming to same guidelines as manuscript figures (noted above)
- If images resemble one from a prior publications, the author must seek permissions (to reproduce or adapt) from the original publisher. [You can resubmit your paper while waiting to hear back from the original publisher but please keep us updated]
- All authors must complete a disclosure form prior to acceptance. A link to complete the form has been sent to all coauthors. Please provide the editorial office with updated email addresses if necessary